JCB Journal of Cell Biology

# The PI(4)P phosphatase Sac2 controls insulin granule docking and release

Phuoc My Nguyen[1], Nikhil R. Gandasi[1] ORCID, Beichen Xie[1] ORCID, Sari Sugahara[1,2], Yingke Xu[3], and Olof Idevall-Hagren[1] ORCID

**Insulin granule biogenesis involves transport to, and stable docking at, the plasma membrane before priming and fusion. Defects in this pathway result in impaired insulin secretion and are a hallmark of type 2 diabetes. We now show that the phosphatidylinositol 4-phosphate phosphatase Sac2 localizes to insulin granules in a substrate-dependent manner and that loss of Sac2 results in impaired insulin secretion. Sac2 operates upstream of granule docking, since loss of Sac2 prevented granule tethering to the plasma membrane and resulted in both reduced granule density and number of exocytic events. Sac2 levels correlated positively with the number of docked granules and exocytic events in clonal β cells and with insulin secretion in human pancreatic islets, and Sac2 expression was reduced in islets from type 2 diabetic subjects. Taken together, we identified a phosphoinositide switch on the surface on insulin granules that is required for stable granule docking at the plasma membrane and impaired in human type 2 diabetes.**

## Introduction

Proinsulin is packaged into granules that bud off from the trans-Golgi network and undergo a series of maturation steps that include maturation of the cargo and alterations in the granule protein and lipid composition. Mature granules dock at the plasma membrane, where they await signals carrying the instructions for fusion (Röder et al., 2016). A typical β cell contains 10,000 granules, but <100 of these are fusion competent and docked at the plasma membrane. Prolonged stimulation of insulin secretion requires replenishment of this pool, and this process mainly involves the recruitment of newly formed granules, highlighting the importance of continuous insulin granule biogenesis for the normal secretory function of β cells (Hou et al., 2009). In type 2 diabetes, defects in insulin granule docking at the plasma membrane result in reduced numbers of fusion-competent granules and contribute to the impaired insulin secretion associated with this disease (Gandasi et al., 2018). The specific steps underlying insulin granule maturation, trafficking, and docking are not well characterized but involve the action of numerous small GTPases of the Rab family and their effector proteins. Constitutive secretion, which is a much better characterized process than the regulated secretion of insulin, also involves the sequential action of specific Rab GTPases. These act in concert with phosphoinositide lipids to recruit effector proteins that promote granule transport and the acquisition of key factors for exocytosis (De Matteis et al., 2005). The

trans-Golgi is rich in the phosphoinositide phosphatidylinositol 4-phosphate (PI(4)P), and this lipid is also required for the formation of Golgi-derived transport vesicles (Cruz-Garcia et al., 2013; De Matteis et al., 2013). The presence of PI(4)P on the newly formed secretory vesicles has been demonstrated in yeast, and it is thought that mammalian cells share this property (Santiago-Tirado et al., 2011). Insulin granules have a very high phosphatidylinositol content, but the relative abundance of its phosphorylated derivatives is not known (MacDonald et al., 2015). In yeast, PI(4)P plays a crucial role in vesicle maturation by promoting myosin-dependent granule transport (Santiago-Tirado et al., 2011) and recruiting the Rab guanine exchange factor Sec2p that in turn activates the Rab GTPase Sec4 and binds the exocyst component Sec15 (Mizuno-Yamasaki et al., 2010). The latter step require removal of PI(4)P, and in yeast, this depends on interactions between PI(4)P and the lipid transport protein Osh4p (Ling et al., 2014). In addition to a putative direct role of Osh4p in PI(4)P transport (de Saint-Jean et al., 2011), it has also been suggested that Osh4p recruits the ER-localized PI(4)P phosphatase Sac1p, leading to the conversion of PI(4)P into phosphatidylinositol (Ling et al., 2014). It is not known if a similar mechanism exists for regulated secretion.

PI(4)P dephosphorylation in mammalian cells is catalyzed by numerous PI(4)P phosphatases (Guo et al., 1999; Foti et al., 2001; Rohde et al., 2003; Hsu et al., 2015; Nakatsu et al., 2015). Sac1 is

[1]Department of Medical Cell Biology, Uppsala University, Uppsala, Sweden; [2]Laboratory of Health Chemistry, Graduate School of Pharmaceutical Sciences, University of Tokyo, Tokyo, Japan; [3]Department of Biomedical Engineering, Key Laboratory of Biomedical Engineering of Ministry of Education, Zhejiang Provincial Key Laboratory of Cardio-Cerebral Vascular Detection Technology and Medicinal Effectiveness Appraisal, Zhejiang University, Hangzhou, China.

Correspondence to Olof Idevall-Hagren: olof.idevall@mcb.uu.se.

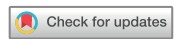

ubiquitously expressed and required to keep low PI(4)P levels in the ER (Foti et al., 2001; Zewe et al., 2018). Sac2/INPP5F is a recently characterized, predominantly neuronal PI(4)P phosphatase that localizes to endosomes and participate in endosome maturation, receptor recycling, and phagocytosis (Hsu et al., 2015; Nakatsu et al., 2015; Levin et al., 2017). We now report that Sac2 is highly expressed in cells of the endocrine pancreas, where it localizes not only to early endosomes but also to insulin granules. Loss of Sac2 resulted in impaired insulin granule docking, leading to reduced granule density at the plasma membrane and impaired insulin secretion. We also found that Sac2 mRNA levels are reduced in pancreatic islets from human donors with type 2 diabetes.

## Results

### Insulin granule PI(4)P dephosphorylation augments insulin secretion

To determine to what extent phosphoinositide contributes to the release competence of insulin granules, we used a light-induced dimerization system to acutely recruit phosphoinositide-metabolizing enzymes to the surface of insulin granules and measured the impact on secretion. The blue-light receptor CRY2 was fused to mCherry and a phosphoinositide-metabolizing enzyme and coexpressed with GFP-tagged CIBN (N-terminal region of cryptochrome-interacting basic-helix-loop-helix) fused to the insulin granule marker Rab3a in clonal insulin-secreting MIN6 β cells. Blue-light illumination resulted in immediate immobilization of CRY2 fusion proteins on Rab3a-positive granules (Fig. 1, A–C). Next, we assessed the impact of recruiting the 4′-phosphatase domain of Sac1 (PI(4)P→PI), the 5′-phosphatase domain of OCRL (phosphatidylinositol 4,5-bisphosphate [PI(4,5)$P_2$]→PI(4)P), or the iSH2-domain of the p85 regulatory subunit of phosphoinositide 3 (PI3) kinase (PI(4,5)$P_2$→phosphatidylinositol 3,4,5 trisphosphate) on insulin secretion using growth hormone (GH) as a proxy for insulin to selectively measure release from transfected cells. We found that secretion stimulated by 20 mM glucose was increased by 65 ± 30% ($P < 0.05$, $n = 3$) following recruitment of the 4′-phosphatase domain to insulin granules but unaffected by the recruitment of the 5′-phosphatase domain or PI3 kinase (Fig. 1 D). We confirmed the substrate specificity of the domains used by recruiting them to the plasma membrane using CIBN-CAAX as bait while measuring the redistribution of PI(4)P (mCh-P4M$_{SidM}$), PI(4,5)$P_2$ (mRFP-PH$_{PLCδ1}$), and phosphatidylinositol 3,4,5 trisphosphate (mRFP-PH$_{Grp1}$) biosensors by total internal reflection fluorescence (TIRF) microscopy. We also ensured that the recruitment to granules was without effect on plasma membrane phosphoinositides (Fig. S1).

### Sac2 is present on insulin granules

Of the putative 4′-phosphatases that may catalyze PI(4)P dephosphorylation on the granule surface, Sac2 is the most attractive candidate, since it has been shown to localize to vesicular structures and be highly expressed in neuronal and pancreatic tissues (Hsu et al., 2015; Nakatsu et al., 2015). Analysis of the expression profile of Sac2 in human tissues

(Human Proteome Atlas) confirmed significantly higher expression in nervous tissues compared with all other tissues examined. We next performed RT-PCR analysis on mouse tissues and found that the Sac2 expression level in pancreatic islets was comparable to that in the brain, indicating important roles in the islet cell types (Fig. 1, E and F). Next, we expressed GFP-tagged Sac2 in MIN6 β cells and imaged the cells by both TIRF and confocal microscopy. We observed numerous weak, small punctate structures that were either static or dynamic (Fig. S2). A previous study demonstrated that inactivation of the 4′-phosphatase domain of Sac2 results in a stronger punctate distribution (Hsu et al., 2015). In line with this, a phosphatase-dead version of Sac2 (mCherry-Sac2-DN) showed more pronounced punctate localization that overlapped with wild-type Sac2 and the early endosomal marker GFP-Rab5 as well as markers of insulin granules (GFP-Rab3a, GFP-Rab27a, and NPY-GFP). Very little overlap between mCherry-Sac2-DN and the lysosomal marker GFP-LAMP1 was observed (Fig. 1, G–I; and Fig. S2). Given that we did not observe any colocalization between the NPY-mCherry–labeled granules and markers of the endocytic compartment (Rab5, Rab4, and Rab10; Fig. S2), these results show that Sac2 localizes to both the endo- and exocytic pathways and suggest a putative role of Sac2 in the regulation of exocytosis.

### Sac2 binding to insulin granules depends on PI(4)P and Rab3

Sac2 contains a conserved Sac phosphatase domain and a homology Sac2 domain. The Sac domain shows a high selectivity for PI(4)P in vitro, whereas the homology Sac2 domain has important functions in Sac2 dimerization and intracellular localization (Hsu et al., 2015). To confirm the substrate specificity of Sac2, we generated optogenetic modules of this enzyme. In brief, the N-terminal 590 amino acids (containing the phosphatase domain) were fused to the blue-light receptor CRY2 (CRY2-Sac2$_{1–590}$), while the binding partner of CRY2 was targeted to the plasma membrane by a CAAX motif (CIBN-CAAX). Confocal and TIRF imaging of cells expressing these fusion proteins showed that CRY2-Sac2$_{1–590}$ was mostly cytosolic. Blue-light illumination resulted in the rapid redistribution of CRY2-Sac2$_{1–590}$ to the plasma membrane, resulting in immediate disappearance of the PI(4)P reporter mCherry-P4M$_{SidM}$ from the same compartment (Fig. 2, A and B). Blue-light illumination was without effect on CRY2-Sac2$_{1–590}$ cellular distribution and PI(4)P levels when CIBN-CAAX was not coexpressed (Fig. 2, C and E). A phosphatase-dead version of CRY2-Sac2$_{1–590}$ (CRY2-Sac2-DN) was readily recruited to the plasma membrane but was without effect on PI(4)P levels (Fig. 2, D and E). These results show that Sac2 has PI(4)P phosphatase activity in situ. We next tested if the localization onto granules depended on the presence of PI(4)P on the granule surface. Wortmannin (10 µM), a nonspecific type III PI4-kinase inhibitor, had little effect on GFP-Sac2-DN binding to insulin granules. Similar results were obtained with the type IIIβ PI4-kinase–selective inhibitor PIK93 (30 µM). Phenylarsine oxide (PAO; 50 µM), which primarily targets type IIIα PI4 kinase, did partially prevent GFP-Sac2-DN binding to insulin granules (Fig. 2, F, G, and J; and Fig. S3). Type II PI4 kinases can localize to vesicular membranes and are insensitive to pharmacological inhibition (Wiedemann et al., 1996; Balla

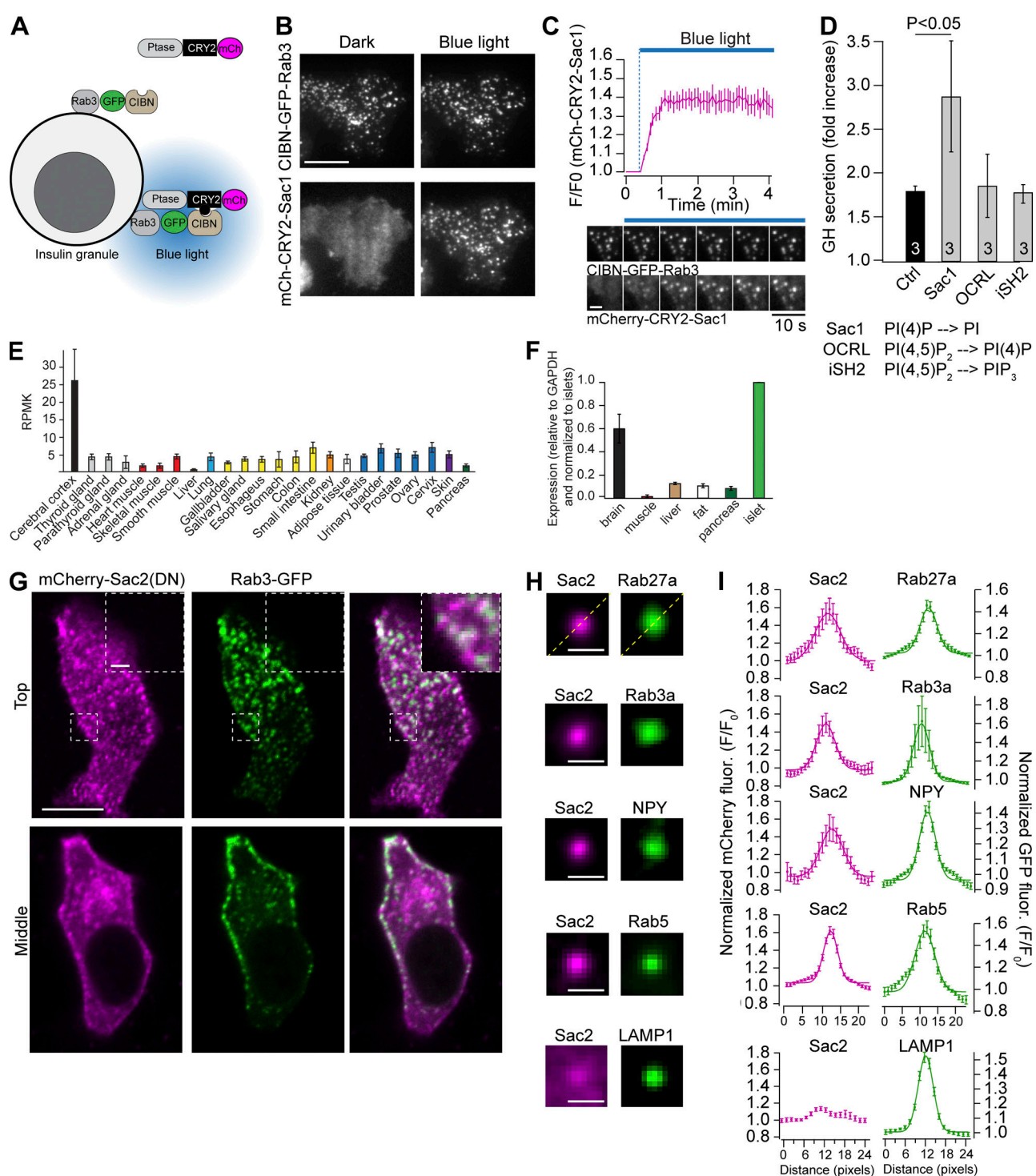

Figure 1. **Sac2 is a component of insulin granules. (A)** Principle of blue-light recruitment of phosphoinositide-metabolizing enzymes onto insulin granules. In the absence of blue light, the CRY2 fusion protein (magenta) is cytosolic, but upon illumination, it binds to CIBN and becomes immobilized on the surface of Rab3a-positive granules (green). **(B)** TIRF micrographs of MIN6 cells showing light-induced recruitment of mCh-CRY2-4ptase$_{Sac1}$ onto CIBN-GFP-Rab3–positive granules. Scale bar is 10 µm. **(C)** TIRF microscopy recording (top) and images (bottom) showing the time course of mCherry-CRY2-4ptase$_{Sac1}$ recruitment onto CIBN-GFP-Rab3–positive insulin granules (mean ± SEM, $n$ = 11 cells). Scale bar is 2 µm. **(D)** Measurements of GH secretion from MIN6 cells expressing GH and GFP-CRY2-4ptase$_{Sac1}$ (control), GFP-CRY2-4ptase$_{Sac1}$ and CIBN-Rab3a (Sac1), GFP-CRY2-5ptase$_{OCRL}$ and CIBN-Rab3 (OCRL), or GFP-CRY2-iSH2 and CIBN-Rab3 (iSH2). GH is co-released with insulin and used as a proxy to selectively detect secretion from transfected cells. Data are presented as mean ± SEM for three experiments and statistical testing was performed with a two-tailed paired Student's $t$ test. **(E)** Sac2 mRNA expression in human tissues. RNA-sequencing data were obtained from the Human Proteome Atlas (2018–11-02). RPKM, reads per kilobase per million. **(F)** Quantitative RT-PCR measurements of Sac2 mRNA levels in mouse brain, skeletal muscle, liver, adipose tissue, whole pancreas, and pancreatic islets ($n$ = 4 mice; mean ± SEM). **(G)** Confocal micrographs of a MIN6 cell expressing mCherry-Sac2-DN (magenta) and Rab3-GFP (green). Scale bar is 10 µm (inset, 2 µm). **(H)** Average projection images of Rab27a-GFP (225 structures, 12 cells), Rab3a-GFP (212 structures, 11 cells), NPY-GFP (249 structures, 14 cells), Rab5-GFP (200 structures, 10 cells), and LAMP1-GFP (198

structures. 12 cells; green) and the corresponding enrichment of mCherry-Sac2-DN (magenta) at the same site. Scale bar is 0.5 µm. **(I)** Quantifications of the relative enrichment of mCherry-Sac2 (magenta) at organelles labeled with the indicated fluorescent proteins (green). Data are presented as mean ± SEM for line profiles drawn diagonally across the images in F ($n$ = 198–249 structures from ≥10 cells).

et al., 2002). To test if these isoforms were required for the granular localization of Sac2, we reduced the expression of both type IIα and IIβ PI4 kinases (PI4K2A/B) with siRNA. However, reduced expression of type II PI4 kinase was without effect on GFP-Sac2-DN binding to insulin granules (Fig. 2, H–J; and Fig. S3). Together, these results indicate that the association of Sac2 with insulin granules is largely independent of PI4-kinase activity. To more directly investigate the role of PI(4)P in Sac2 granule binding, we used the light-regulated dimerization system to recruit the 4′-phosphatase domain of Sac1 (CRY-2-4′-ptase$_{Sac1}$) to Rab3a-positive granules (CIBN-Rab3) while monitoring the distribution of mCherry-Sac2 (Fig. 2 K). We found that dephosphorylation of granular PI(4)P resulted in 23 ± 3% decrease (P < 0.001, $n$ = 16) in mCherry-Sac2 structures, mostly reflecting reduced association between mCherry-Sac2 and Rab3a-positive insulin granules (Fig. 2, L–O). Sac2 binding to early endosomes depends on interactions with Rab5 (Nakatsu et al., 2015). To test if Rab3 plays a similar role on granules, we coexpressed a GDP-locked form of Rab3a (T35N; GFP-Rab3-DN) and determined its impact on Sac2 binding to insulin granules. Consistent with a role of Rab3a in targeting Sac2 to granules, we found a 40 ± 7% reduction (P < 0.05, $n$ = 11) in Sac2 granular accumulation in cells coexpressing Rab3a-DN (Fig. 2, P and Q). Together, these results show that Sac2 localization to granules requires interactions with both PI(4)P and Rab3a.

### Sac2 is required for glucose-stimulated insulin secretion

The insulin granule localization of Sac2 is consistent with a role of this phosphatase in the control of insulin secretion. To test this possibility, we transiently and stably reduced Sac2 expression in MIN6 β cells using siRNA and shRNA, respectively. Knockdown (KD) efficiency varied between 40% and 60% reduction at the mRNA level, as determined by quantitative RT-PCR (Fig. S4). Attempts to detect Sac2 at the protein level using commercial antibodies failed, similar to what has been reported in other studies (Hsu et al., 2015; Nakatsu et al., 2015). We were, however, able to detect strong suppression of GFP-Sac2 protein levels using fluorescence microscopy and immunoblotting with an anti-GFP antibody (Fig. S5). Confocal microscopy examination showed that the PI(4)P marker GFP-PH$_{OSBP}$ was enriched at insulin granules following both transient and stable KD of Sac2, indicating that Sac2 is responsible for PI(4)P dephosphorylation on insulin granules (Fig. 3, A–D). Next, we measured insulin secretion from control and Sac2 KD cells. We observed no difference in the basal secretion occurring at a low glucose concentration (3 mM) but 41 ± 8% (P < 0.01, $n$ = 6) suppression of the secretion at 20 mM glucose (Fig. 3 E). Similar results were obtained following stable Sac2 KD (Fig. S4). Measurements of insulin content in the cells revealed no difference between control and Sac2 KD cells, indicating a primary defect in insulin secretion and not insulin production (Fig. 3 F).

### Sac2 deficiency impairs depolarization-triggered secretion but lacks effect on compensatory endocytosis

Next, we directly tested the involvement of Sac2 in depolarization-triggered insulin secretion. Control or Sac2 KD MIN6 β cells expressing the pH-sensitive exocytosis marker VAMP2-pHluorin were observed by TIRF microscopy following depolarization with 30 mM K$^+$. Depolarization triggered the fusion of insulin granules with the plasma membrane, seen as bright flashes of light occurring as the pH of the granule lumen was neutralized. The continuous secretion resulted in a progressive accumulation of fluorescent VAMP2 in the plasma membrane that plateaued after ~5 min, indicating that the rates of exocytosis and endocytosis were the same. In cells with transient or stable Sac2 KD, the accumulation of VAMP2 in the plasma membrane was significantly reduced in response to depolarization, indicating a secretory defect (Fig. 4, A and B). This response could be normalized by reexpression of wild-type Sac2, but not a catalytically inactive mutant (Fig. 4, B and C). Taking advantage of cells with variable degree of Sac2 KD efficiency, we next compared the depolarization-induced secretory response and found that this correlated positively with Sac2 expression levels ($R^2$ = 0.954, P < 0.01; Fig. 4 D). The depolarization-induced Ca$^{2+}$ influx, measured with Fluo-4, was unaffected by Sac2 KD, showing that the impaired secretion in these cells occurs independent of changes in cytosolic Ca$^{2+}$ concentrations (Fig. 4 E). In a previous study, catalytically inactive Sac2 (Sac2-DN) was found to operate in a dominant-negative manner, thereby phenocopying the KD (Hsu et al., 2015). In line with this observation, we found that overexpression of Sac2-DN, but not wild-type Sac2, reduced the VAMP2 plasma membrane insertion to the same extent as Sac2 KD, emphasizing the need of the 4′-phosphatase activity for normal insulin secretion (Fig. 4 F). Sac2 was previously described as a regulator of ligand-triggered endocytosis and phagocytosis (Hsu et al., 2015; Nakatsu et al., 2015; Levin et al., 2017). To test whether it might play a similar role in β cells, we compared the rate of VAMP2 reuptake following removal of the depolarizing stimulus and found that there was no difference between control or Sac2 KD cells, while reexpression of wild-type Sac2 in Sac2 KD cells moderately increased the rate of membrane reuptake (Fig. 4, G and H). Together, these results show that Sac2 4′-phosphatase activity is needed for normal Ca$^{2+}$-triggered exocytosis of insulin granules, but not for the compensatory reuptake of membrane.

### Sac2 is required for insulin granule docking to the plasma membrane

A possible explanation for the observed defect in insulin secretion in cells with reduced Sac2 expression could be alterations in granule fusion with the plasma membrane. We therefore imaged the depolarization-induced fusion of individual NPY-mCherry–positive insulin granules by TIRF microscopy. We found that the kinetics of granule fusion was unaffected by Sac2 KD, but the number of fusion

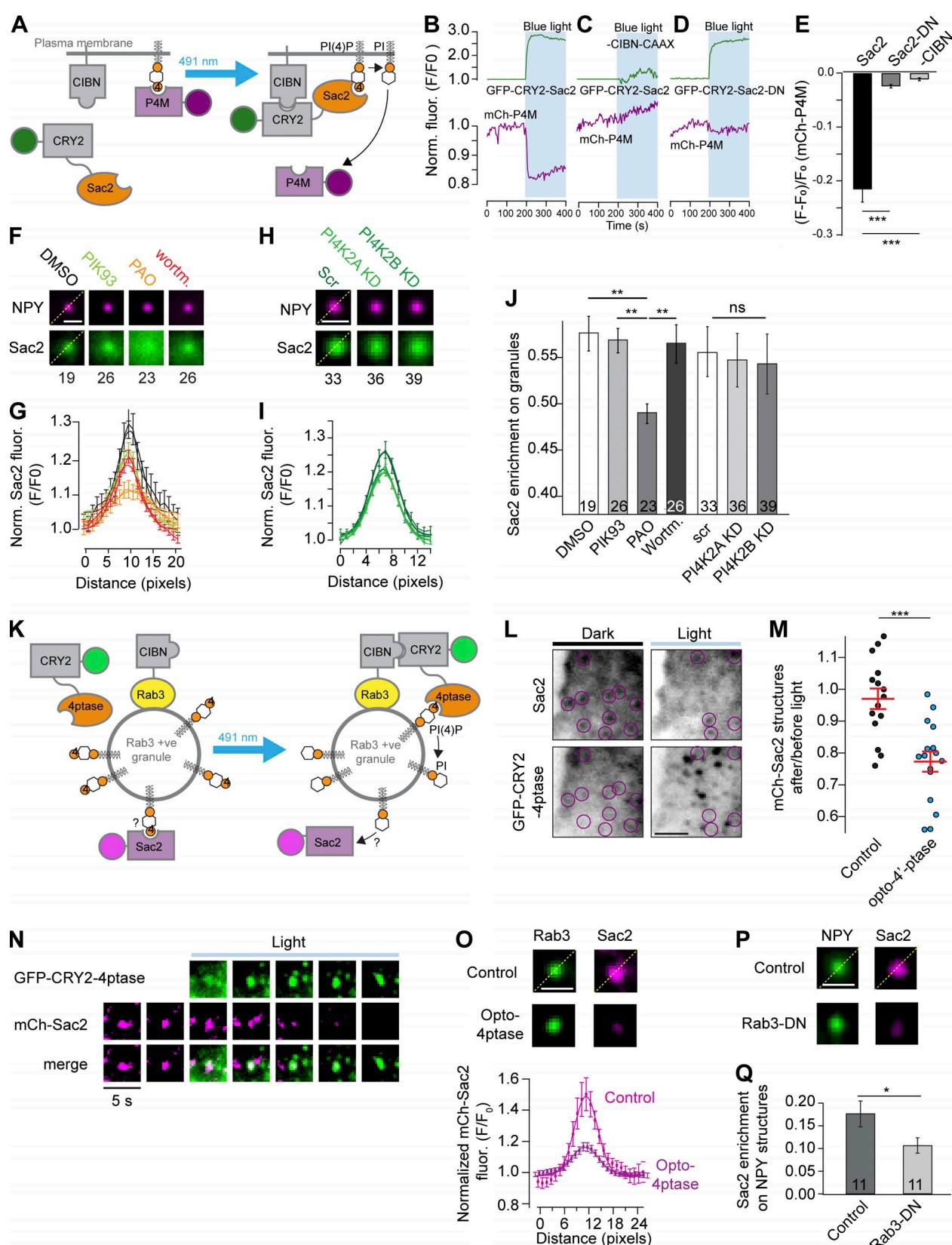

Figure 2. **Sac2's association with insulin granules requires PI(4)P and Rab3. (A)** Principle of blue-light–dependent recruitment of the 4'-phosphatase domain of Sac2 to the plasma membrane based on CRY2-CIBN interaction and detection of changes in plasma membrane PI(4)P levels using mCherry-P4M$_{SidM}$. **(B–D)** TIRF microscopy recordings from single MIN6 cells expressing the PI(4)P marker mCh-P4M$_{SidM}$ together with CIBN-CAAX and GFP-CRY2-Sac2$_{1–590}$ (B), GFP-CRY2-Sac2$_{1–590}$ only (C), or CIBN-CAAX and GFP-CRY2-Sac2-DN (D) before and during blue-light illumination. **(E)** Mean ± SEM mCh-P4M$_{SidM}$ dissociation

from the plasma membrane under the experimental conditions shown in B–D; *n* = 20 (B), 12 (C), and 8 (D) cells. ***, P < 0.001, two-tailed Student's unpaired *t* test. **(F and G)** Line profiles showing the enrichment of GFP-Sac2-DN on NPY-mCherry–positive structures in the presence of DMSO (1 µl/ml, *n* = 19 cells), PIK93 (30 µM, *n* = 26 cells), PAO (50 µM, *n* = 23 cells), and wortmannin (10 µM, *n* = 26 cells). Data are presented as mean ± SEM. Images in F are average projections of >20 NPY-positive granules per cell. Scale bar is 1 µm. **(H and I)** Line profiles showing the enrichment of GFP-Sac2-DN on NPY-mCherry–positive structures in MIN6 cells treated with control siRNA (*n* = 33 cells) or siRNA targeting PI4K2A (*n* = 36 cells) or PI4K2B (*n* = 39 cells). Data are presented as mean ± SEM. Images in H are average projections of >20 NPY-positive granules per cell for the indicated number of cells. **(J)** Quantifications of the GFP-Sac2-DN enrichment at NPY-mCherry–positive structures after the indicated treatments. Numbers at the bottom indicate the number of cells observed. **, P < 0.01, two-way ANOVA with Tukey's post hoc test. See Materials and methods for details on how the enrichment was estimated. **(K)** Principle of blue-light–dependent recruitment of the 4′-phosphatase domain of Sac1 to the insulin granule based on the CRY2–CIBN interaction and detection of changes in granule-associated mCherry-Sac2-DN fluorescence. **(L)** TIRF micrographs of a MIN6 cell expressing mCherry-Sac2-DN, CIBN-Rab3, and GFP-CRY2-4ptase$_{Sac1}$ before and 2 min after blue-light illumination. The position of mCherry-Sac2–positive structures are indicated by magenta circles, and images have been inverted to show fluorescence in black. Scale bar is 2 µm. **(M)** Quantification of the density of mCherry-Sac2-DN structures before and 2 min after blue-light illumination in control cells (CIBN-Rab3, 14 cells) and cells expressing CIBN-Rab3 and GFP-CRY2-4ptase$_{Sac1}$ (16 cells). ***, P < 0.001, two-tailed unpaired Student's *t* test. **(N)** TIRF micrographs showing light-induced recruitment of GFP-CRY2-4ptase$_{Sac1}$ onto a Rab3-positive granule with the corresponding loss of mCherry-Sac2-DN. **(O)** Average projections of Rab3-CIBN-GFP–labeled insulin granules and the corresponding accumulation of mCherry-Sac2-DN in control cells and in cells coexpressing CRY2-4ptase$_{Sac1}$ (Opto-4ptase) following 2 min of blue-light illumination (*n* = 150 structures from 10 cells). Quantifications of the relative enrichment of mCherry-Sac2 at CIBN-GFP-Rab3 structures are shown below (mean ± SEM). Scale bar is 1 µm. **(P)** Average projections of NPY-GFP–labeled insulin granules and the corresponding accumulation of mCherry-Sac2 in control and Rab3-DN–overexpressing cells (*n* = 165 structures from 11 cells). Scale bar is 1 µm. **(Q)** Quantifications of the enrichment of mCherry-Sac2 on NPY-GFP–labeled granules in control cells and in cells expressing Rab3-DN (mean ± SEM for 11 cells) *, P < 0.05, two-tailed unpaired Student's *t* test.

events was reduced by 70% (Fig. 5, A and B). Examination of the granule density at the plasma membrane revealed 51% reduction in Sac2 KD cells, which could be normalized by the re-expression of wild-type, but not catalytically inactive, mCherry-Sac2 (Fig. 5, C and D). Given that there was no difference in insulin content in Sac2 KD cells, the reduction in granule density could be the result of reduced ability of granules to stably dock at the plasma membrane. To test this, we imaged individual insulin granule docking events in control and Sac2 KD cells. We found that although a similar number of granules approached the plasma membrane in both control and Sac2 KD cells, seen as their appearance within the evanescent field of the TIRF microscope, <10% of the granules became stably docked in the Sac2 KD cells compared with 50% in the control cells (Fig. 5 E). Insulin granule docking to the plasma membrane involve interactions between proteins and lipids in the respective membrane. Class II PI3 kinase and its product, phosphatidylinositol 3-phosphate (PI(3)P), are present on the secretory granules of PC-12 cells (Meunier et al., 2005; Wen et al., 2008). To test the hypothesis that Sac2 provides substrate for a granule-localized PI3 kinase, we examined the distribution of the PI(3)P sensor GFP-2xFYVE$_{EEA1}$ in control and Sac2 KD cells by confocal microscopy. We could not observe any accumulation of PI(3)P on the surface of insulin granules in either cell line (Fig. 6 A). To further test if acute dephosphorylation of PI(4)P on the granule surface results in rapid PI(3)P synthesis, we quantified the amount of GFP-2xFYVE$_{EEA1}$ present on insulin granules before and after light-induced recruitment of the 4′-phosphatase domain of Sac1 to the granule surface. This manipulation, however, did not induce GFP-2xFYVE$_{EEA1}$ accumulation at the granule surface, speaking against the involvement of PI3 kinase downstream of Sac2 recruitment (Fig. 6, B–D). Phosphatidylserine (PS) is another insulin granule lipid that plays a role in granule release (Sánchez-Migallón et al., 1994; MacDonald et al., 2015). PS accumulation in the plasma membrane is at least partly due to the action of the oxysterol-binding protein (OSBP)–related proteins ORP5 and ORP8 that exchange PS for PI(4)P (Chung et al.,

2015). To test if granule PI(4)P might drive PS accumulation we examined the distribution of the PS-binding protein GFP-LactC2 (Yeung et al., 2008) in control and Sac2 KD cells. In control cells, we found strong accumulation of the PS sensor on the surface of both docked insulin granules and granules located deeper in the cell (Fig. 6, E and F). Depolarization of these cells resulted in exocytosis, seen by TIRF microscopy as a rapid disappearance of NPY-mCherry–positive granules that was accompanied by a somewhat slower disappearance of GFP-LactC2 fluorescence from the same compartment, consistent with lateral diffusion of PS in the plasma membrane following granule fusion (Fig. 6, H–J). The cellular distribution of GFP-LactC2 and its enrichment on insulin granules was unaffected by Sac2 KD, indicating that granule PI(4)P is not involved in the accumulation of PS (Fig. 6 G). Besides lipids, numerous proteins are important determinants of insulin granule docking, including Rab3a and Rab27a, as well as their effectors. However, confocal microscopy examination of the distribution of Rab3a, Rab27a, and granuphilin did not show any difference between control cells and cells with reduced Sac2 expression (Fig. S5).

## Sac2 expression is decreased in type 2 diabetes

Impaired docking was recently described to occur in β cells from type 2 diabetics and contribute to the secretory defect observed in these cells (Gandasi et al., 2018). To investigate whether Sac2 might play a role in this docking defect, we assessed Sac2 mRNA expression levels using previously published RNA-sequencing datasets (Fadista et al., 2014; Gandasi et al., 2018) and correlated these to the stimulatory index, a measure of insulin secretion capacity, of healthy donors. Interestingly, we found a positive correlation between Sac2 expression and secretion, similar to what we observed in the cell lines with different Sac2 mRNA levels (Fig. 7 A). Moreover, we found that Sac2 expression was significantly reduced in pancreatic islets from patients with type 2 diabetes when compared with nondiabetics, suggesting that Sac2 deficiency might contribute to disease development or progression (Fig. 7 B). To experimentally test this hypothesis, we overexpressed GFP-Sac2 and determined the

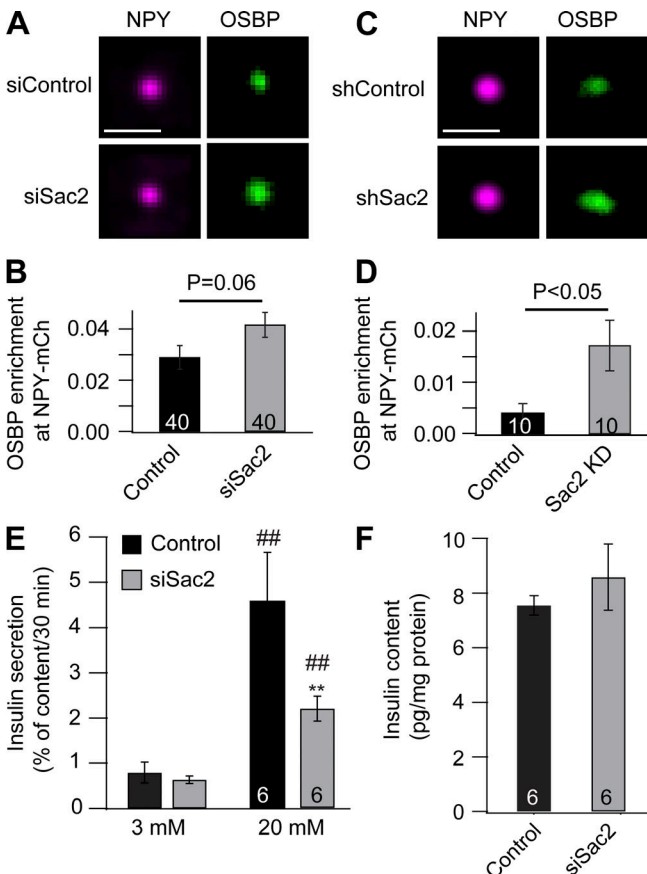

**Figure 3. Sac2 is required for normal glucose-stimulated insulin secretion. (A)** Average projection images showing the colocalization of GFP-PH$_{OSBP}$ with NPY-mCherry in cells treated with control and Sac2 siRNA (640 structures from 40 cells). Scale bar is 1 µm. **(B)** Quantification of the enrichment of GFP-PH$_{OSBP}$ on NPY-mCherry–positive granules in cells treated with control or Sac2 siRNA (mean ± SEM for the indicated number of cells; statistical analysis was performed using a two-tailed unpaired Student's $t$ test). **(C)** Average projection images showing the colocalization of GFP-PH$_{OSBP}$ with NPY-mCherry in control and Sac2 KD cells (200 structures from 10 cells). Scale bar is 1 µm. **(D)** Quantification of the enrichment of GFP-PH$_{OSBP}$ on NPY-mCherry–positive granules in control and Sac2 KD cells (mean ± SEM for the indicated number of cells; statistical analysis was performed using a two-tailed unpaired Student's $t$ test). **(E)** Insulin secretion measurements from MIN6 cells transfected with control or Sac2 siRNA. Cells were incubated for 30 min in 3 mM or 20 mM glucose. Data are presented as mean ± SEM for six separate experiments (##, P < 0.01 versus 3 mM; **, P < 0.01 versus 20 mM control; two-tailed paired Student's $t$ test). **(F)** Measurements of insulin content from the cells used in E.

## Discussion

We show here that the previously characterized endocytic factor Sac2 also plays a prominent role in the regulation of cargo release. We found that Sac2 associates with insulin granules and that this association, with accompanying dephosphorylation of PI(4)P, is required for stable granule docking at the plasma membrane. Consequently, loss of Sac2 resulted in reduced granule density at the plasma membrane with accompanying loss of insulin secretion. We also found that the expression level of Sac2 positively correlated with insulin secretion in both cell lines and human pancreatic islets and that the expression of Sac2 was reduced in islets isolated from patients with type 2 diabetes.

The Golgi is rich in PI(4)P, mainly due to the activity of phosphatidylinositol 4-kinase IIIβ, and many Golgi-localized proteins are equipped with PH domains that allows them to recognize this lipid (De Matteis et al., 2005). PI(4)P is required for the recruitment of budding factors to the trans-Golgi and the formation of secretory granules (Cruz-Garcia et al., 2013). Under conditions of growth deprivation, the ER-localized PI(4)P phosphatase Sac1 redistributes to the Golgi, where it removes PI(4)P and blocks secretion (Faulhammer et al., 2007; Blagoveshchenskaya and Mayinger, 2009), further highlighting the importance of this lipid in the formation of secretory granules. The fate of PI(4)P following granule formation is less clear. Using biosensors for detecting PI(4)P, we were unable to see any strong enrichment of these probes on the surface of insulin granules. This is in line with many previous studies concluding that these sensors are unable to detect all PI(4)P pools and should not been seen as evidence for lack of this lipid on granules (Wills et al., 2018). In fact, phosphatidylinositol, the precursor of PI(4)P, was recently found to constitute ∼20% of all lipids on the surface of insulin granules (MacDonald et al., 2015). Consistent with the presence of PI(4)P on insulin granules, we now show that the PI(4)P phosphatase Sac2 localizes to insulin granules. The localization required binding to PI(4)P, since the granular localization was much more apparent when the enzyme was rendered catalytically inactive, and its association with granules was reduced when PI(4)P was acutely removed from the granule surface. Pharmacological inhibition or siRNA-mediated silencing of PI4 kinases partially prevented the association of Sac2 with insulin granules, indicating that part of the granule PI(4)P is synthesized after budding from the Golgi. The specific PI4-kinase isoforms involved remains to be determined. In addition, we also found that the granular localization depended on interactions with Rab3a, which is analogous to how Sac2 associates with early endosomes (Nakatsu et al., 2015). Coincidence detection of Rabs and PIs are well described for other phosphoinositide-metabolizing enzymes and may help Sac2 gain specificity in site of action (Di Paolo and De Camilli, 2006). Using optogenetics to recruit a 4′-phosphatase to Rab3-positive insulin granules, we found that dephosphorylation of PI(4)P resulted in increased insulin secretion. This observation indirectly confirms the presence of PI(4)P on the granule surface and indicates that PI(4)P removal is part of granule maturation. The localization of Sac2 to insulin granules make this enzyme an attractive candidate for catalyzing this reaction.

Sac2 was previously shown to control phosphoinositide turnover in the endocytic compartment, regulating recycling of integrins and transferrin receptors as well as phagocytosis (Hsu et al., 2015; Nakatsu et al., 2015; Levin et al., 2017). Endocytosis also occurs in parallel with exocytosis both to maintain plasma membrane homeostasis and to serve as a source of new insulin granules (Orci et al., 1973; Ohara-Imaizumi et al., 2002; Wen

impact on insulin granule density. We found that Sac2 overexpression positively correlated with insulin granule density at the plasma membrane, consistent with a direct role of this phosphatase in the control of granule docking (Fig. 7 C).

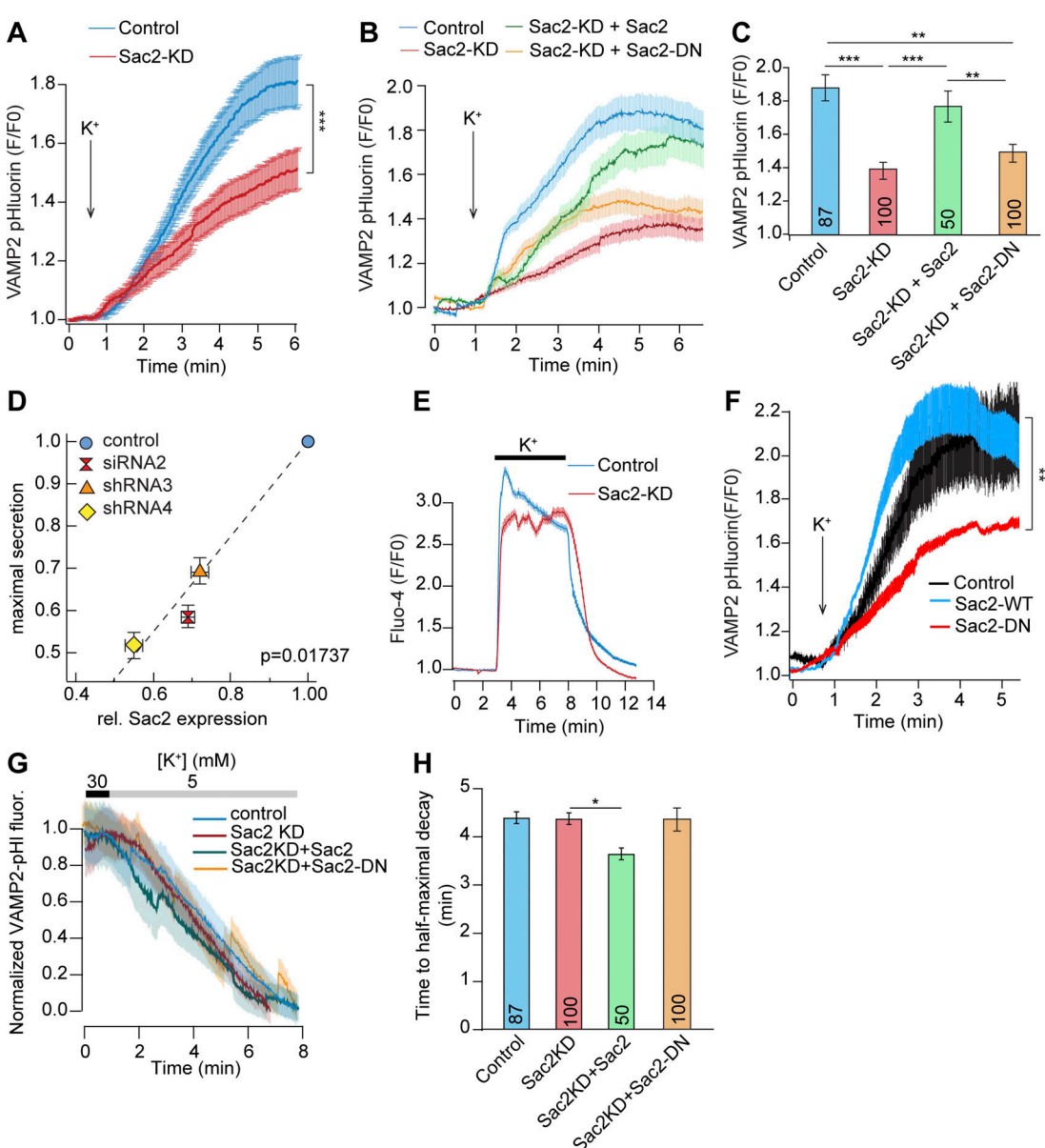

Figure 4. **Sac2 is required for depolarization-induced insulin secretion, but not for compensatory endocytosis. (A)** TIRF microscopy recordings of VAMP2-pHluorin fluorescence change in control and Sac2 KD cells in response to 30 mM K[+] (mean ± SEM for 100 control cells and 100 Sac2 KD cells; ***, P < 0.001, Student's unpaired $t$ test). **(B)** TIRF microscopy recordings of VAMP2-pHluorin fluorescence change in control, Sac2 KD, Sac2 KD (+mCherry-Sac2), and Sac2 KD (+mCherry-Sac2[DN]) cells in response to 30 mM K[+] (mean ± SEM for 87, 100, 50, and 53 cells). **(C)** The maximal VAMP2-pHluorin fluorescence increase in response to depolarization from the cells in B. Values are mean ± SEM; **, P < 0.001; ***, P < 0.0001; Student's unpaired $t$ test). **(D)** Correlation between Sac2 mRNA expression and maximal VAMP2-pHluorin response to 30 mM K[+]. Statistical testing was performed with a linear regression $t$ test. **(E)** TIRF microscopy recordings of Fluo-4 fluorescence change in control and Sac2 KD MIN6 cells in response to 30 mM K[+]. Data are presented as mean ± SEM for 220 (control) and 201 (Sac2-KD) cells. **(F)** TIRF microscopy recordings of VAMP2-pHluorin fluorescence change in cells expressing mCherry (control), mCherry-Sac2, or mCherry-Sac2-DN and stimulated with 30 mM K[+] (mean ± SEM for 27, 66, and 57 cells; **, P < 0.01, Student's unpaired $t$ test). **(G and H)** Quantifications of the reuptake of VAMP2-pHluorin in control, Sac2 KD, Sac2 KD (+mCherry-Sac2), and Sac2 KD (+mCherry-Sac2-DN) cells. Data are presented as mean ± SEM (numbers at the bottom indicate the number of cells observed; *, P < 0.05; Student's unpaired $t$ test).

et al., 2012). Interfering with endocytosis by overexpression of dynamin mutants or dynamin knockout results in impaired insulin secretion (Min et al., 2007; Fan et al., 2015). However, we did not observe any defect in membrane reuptake following stimulation of insulin granule exocytosis in cells with reduced Sac2 expression. This would suggest that Sac2 functions primarily in the secretory compartment of β cells under the experimental conditions tested here. The mechanism by which

Sac2 controls insulin secretion is not entirely clear. We observed reduced insulin granule docking in Sac2 KD cells, and this most likely explains the reduced granule density at the plasma membrane and the impaired secretion. The granules were still able to approach the plasma membrane in Sac2 KD cells but unable to stably associate with it, indicating a defect in granule tethering and not transport. Rab27 and Rab3 are the best-characterized docking factors in β cells (Yaekura et al., 2003;

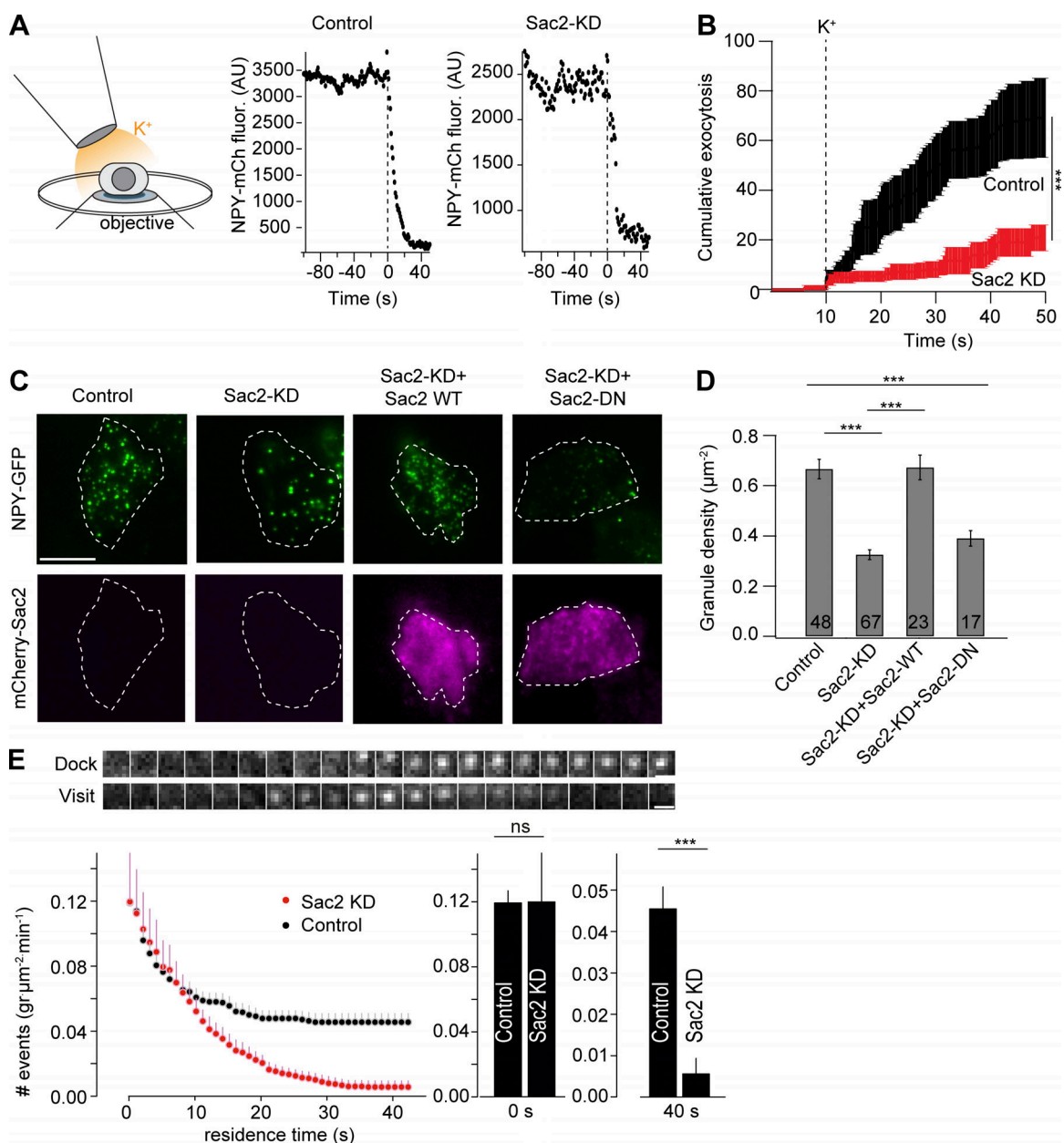

Figure 5. **Sac2 is required for stable granule docking. (A)** Illustration showing the principle for local, acute stimulation of exocytosis from single cells. See Materials and methods for details. Traces showing the kinetics of exocytosis of a single granule from a control and a Sac2 KD cell are presented to the right. **(B)** Quantification of cumulative exocytosis in control and Sac2 KD cells (n = 10 cells for both conditions; ***, P < 0.001, Student's unpaired t test). **(C)** TIRF micrographs showing NPY-GFP granule density at the plasma membrane in control, Sac2 KD, Sac2 KD (+mCherry-Sac2), and Sac2 KD (+mCherry-Sac2-DN) cells. Scale bar is 10 µm. **(D)** Quantifications of NPY-GFP granule density at the plasma membrane for the indicated conditions. Number of analyzed cells are indicated in the figure (mean ± SEM; ***, P < 0.001; Student's unpaired t test). **(E)** Quantifications of NPY-GFP granule (gr.) docking in control and Sac2 KD cells (mean ± SEM for 12 [control] and 10 [KD] cells). Bar graphs show the number of granules that approach the plasma membrane (0 s) and the number of granules that remained stably associated with the plasma membrane 40 s later (40 s) in control and Sac2 KD cells (mean ± SEM). ***, P < 0.001, Student's unpaired t test. Scale bar is 1 µm.

Kasai et al., 2005), but their localization to granules did not require Sac2. The Rab27 effector granuphilin is also required for insulin granule docking (Gomi et al., 2005), but its localization to insulin granules was also Sac2 independent. However, other docking factors, including rabphilin, Exophilin8, Noc2, and Rim2α, remain interesting candidates that may act downstream of Sac2 (Cheviet et al., 2004; Tsuboi and Fukuda, 2005; Yasuda et al., 2010; Fan et al., 2017). These exocytic factors are equipped

with FYVE-like domains similar to the PI(3)P-binding FYVE domains of many endocytic factors. Although PI(3)P has been found on the surface of secretory granules (Wen et al., 2008; Dominguez et al., 2011), we were unable to detect any colocalization between the PI(3)P sensor 2xFYVE$_{EEA1}$ and insulin granules, and acute dephosphorylation of PI(4)P on the granule surface did not result in accumulation of the PI(3)P sensor. It is possible that the PI(3)P sensor used here, which is from an

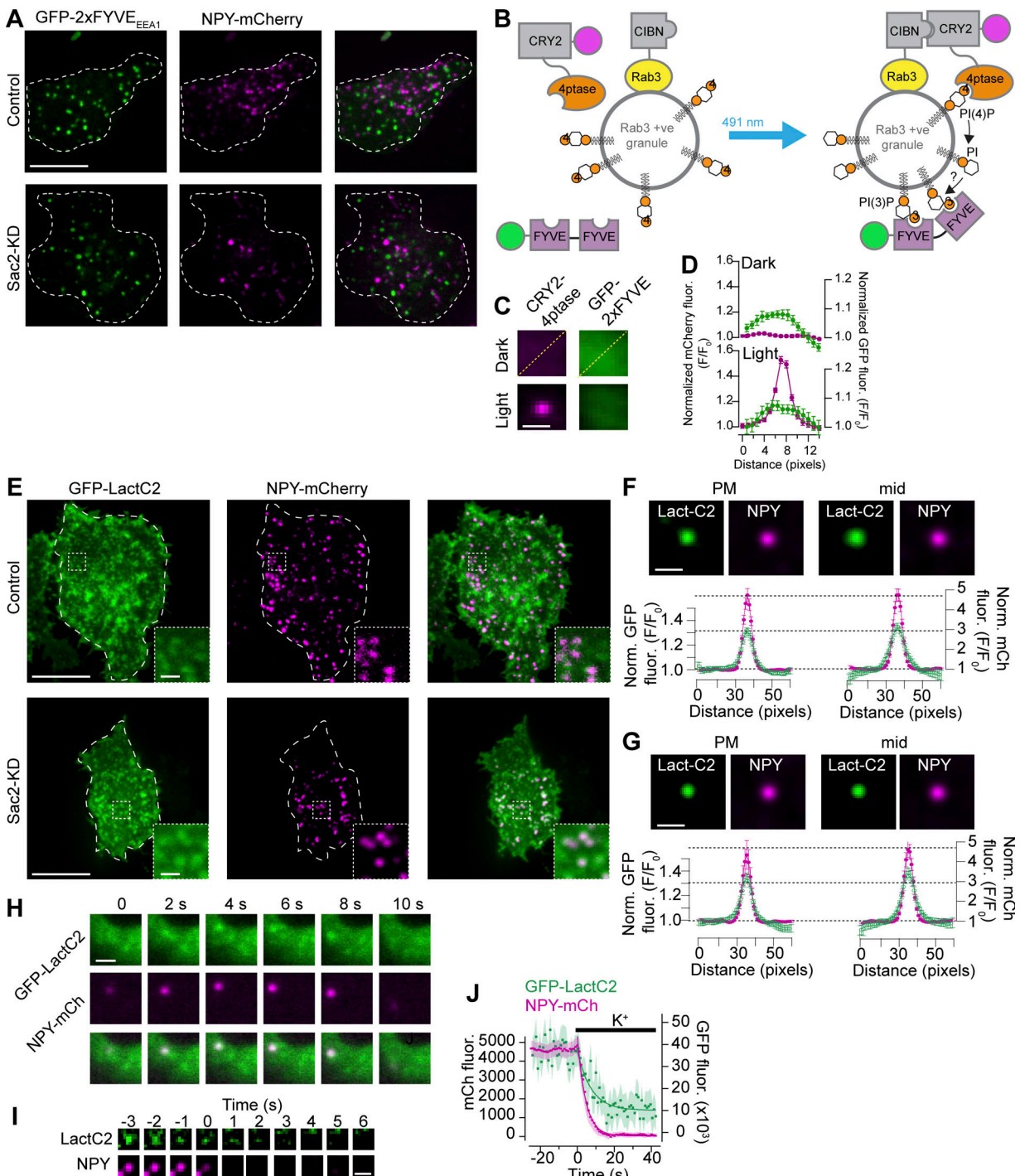

Figure 6. **Sac2 KD is without effect on granule PI(3)P or PS levels. (A)** Confocal microscopy images of control and Sac2 KD MIN6 cells expressing the PI(3)P biosensor GFP-2xFYVE$_{EEA1}$ and the insulin granule marker NPY-mCherry. Scale bar is 10 µm. **(B)** Schematic drawing showing the principle of light-induced recruitment of the 4′-phosphatase domain of Sac1 onto an insulin granule and its impact on granule PI(3)P levels. **(C)** Images show the average mCherry-CRY2-4′-ptase$_{Sac1}$ fluorescence (magenta) and the associated GFP-2XFYVE$_{EEA1}$ fluorescence (green) before and after blue-light illumination. Scale bar is 1 µm. **(D)** Line profiles from lines drawn diagonally across the images in B. Data are presented as mean ± SEM for 200 structures in 10 cells. **(E)** Confocal micrographs of control and Sac2 KD MIN6 cells expressing the PS sensor GFP-LactC2 (green) and the insulin granule marker NPY-mCherry (magenta). Scale bars are 10 µm (inset, 1 µm). **(F and G)** Images show the average NPY-mCherry fluorescence (magenta) and the associated GFP-LactC2 fluorescence (green) from control (F) and Sac2 KD (G) cells. Images from both basolateral (plasma membrane [PM]) and middle (mid) confocal sections are shown. Line profiles below are from lines drawn diagonally across the images. Data are presented as mean ± SEM for 21 (control) and 19 (Sac2 KD) cells (15–20 structures per cells). Scale bars are 1 µm. **(H)** Confocal microscopy images of GFP-LactC2 (green) and NPY-mCherry (magenta) showing that the two proteins colocalize at motile structures. Scale bar is 1 µm. **(I)** TIRF microscopy images of an insulin granule (magenta) and the associated GFP-LactC2 fluorescence (green) during fusion triggered by 90 mM K⁺. Scale bar is 1 µm. **(J)** TIRF microscopy recordings of K⁺-induced insulin granule fusion (NPY-mCherry; magenta) with the plasma membrane and the corresponding change in granule-associated GFP-LactC2 fluorescence (green; mean ± SEM for 12 cells). Curve fitting was done with a single exponential function.

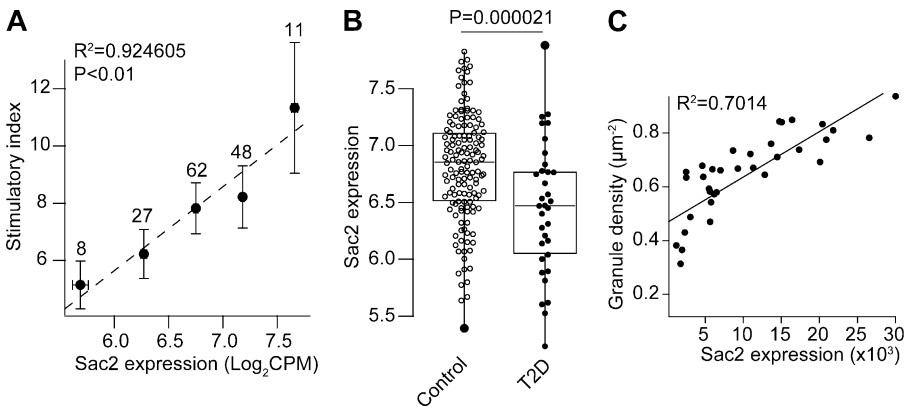

Figure 7. **Sac2 expression is reduced in human type 2 diabetes. (A)** Correlation between stimulatory index and Sac2 mRNA expression in islets from healthy donors (*n* = 156). Data were binned into five groups based on Sac2 expression and are presented as mean ± SEM. **(B)** Sac2 mRNA expression in islets from control and type 2 diabetic (T2D) donors (*n* = 156 control, 34 type 2 diabetics). **(C)** Correlation between mCherry-Sac2 expression and granule density at the plasma membrane (NPY-GFP labeled; *n* = 34 cells).

endocytic factor, fails to recognize the granule pool of PI(3)P. It is also possible that the FYVE domains of the exocytic factors bind to other PIs, as it has been found that not all FYVE domains are PI(3)P selective (Gil et al., 2012).

In addition to being a direct effector or precursor, PI(4)P may also control the lipid composition of granular membranes by recruiting lipid-transport proteins (Ling et al., 2014). The insulin granule membrane is rich in both PS and cholesterol, both of which can be exchanged for PI(4)P at membrane contact sites (Chung et al., 2015; Mesmin et al., 2017). PI(4)P/PS counter-transport at ER–plasma membrane contacts is mediated by ORP5/8, and these proteins have also been found at ER–mitochondria contacts, although the importance of lipid transport at this site is unclear (Galmes et al., 2016). Although physical contacts between the ER and secretory granules have not been observed, it is possible that excess accumulation of granule PI(4)P as a consequence of Sac2 KD could drive such interactions, since they are in part PI(4)P dependent (Chung et al., 2015). We were, however, not able to detect any change in the granule PS content in cells with reduced Sac2 expression, arguing against a role of granule PI(4)P in PS accumulation. An interesting, still unexplored, possibility is that granule PI(4)P is required for cholesterol accumulation in the granule membrane. Insulin granules are rich in cholesterol, and both cholesterol deficiency and overloading results in defect insulin granule biogenesis and secretion (Bogan et al., 2012; Xu et al., 2017; Hussain et al., 2018). OSBP exchanges ER-derived cholesterol for Golgi PI(4)P (Mesmin et al., 2017), and loss of OSBP has been shown to inhibit insulin secretion by preventing the granules from accumulating cholesterol (Hussain et al., 2018). We find that the PH domain of OSBP weakly binds to the insulin granules, an interaction that is enhanced in cells with reduced Sac2 levels. This observation is consistent with OSBP or related proteins acting as PI(4)P sensors on the insulin granule surface. Interestingly, such a mechanism would resemble the proposed model for how secretory vesicle PI(4)P regulates constitutive secretion in yeast (Ling et al., 2014).

It was recently shown that the expression of multiple genes encoding components of the docking machinery, including Rab3a and Rab27a, were reduced in islets from type 2 diabetics and that the overexpression of these components could promote granule docking (Gandasi et al., 2018). Knockout mice lacking Rab3 or Rab27 both exhibit reduced insulin secretion as a consequence of impaired granule docking at the plasma membrane, consistent with the notion that these proteins are involved in the pathogenesis of type 2 diabetes (Yaekura et al., 2003; Kasai et al., 2005). Similarly, we now found that loss of Sac2 resulted in impaired granule docking and release. We also show that the expression level of Sac2 correlates positively with insulin granule docking and exocytosis in clonal β cells and insulin secretion from pancreatic islets from nondiabetic subjects. Moreover, we found that Sac2 expression was reduced in islets from type 2 diabetic subjects. Taken together, these findings are consistent with Sac2 insufficiency being a contributing factor to the pathological changes observed in β cells from type 2 diabetics. These results also show that not only the protein composition of the granules but also the lipid composition is crucial for the normal function of the secretory machinery. Given that phosphoinositides are under metabolic control in β cells (Wuttke et al., 2010; Xie et al., 2016) and that changes in cellular metabolism is a dominating feature of type 2 diabetes, we anticipate that changes in granule lipid composition may be a consequence of both altered transcription, as shown here, as well as metabolic changes in substrate concentrations. Drugs aimed at improving granule docking and restoring granule density at the plasma membrane represent an attractive and unexplored strategy for the development of anti-diabetic drugs.

## Materials and methods
### Reagents
All salts were from Sigma-Aldrich. PAO was from Sigma-Aldrich, and wortmannin and PIK93 were from TOCRIS Bioscience. The following plasmids were used: GFP-PH-OSBP (Balla et al., 2005; gift from Tamas Balla, National Institutes of Health, Bethesda, MD), GFP-PHx2-OSH2 (Stefan et al., 2011; plasmid 36095; Addgene), GFP-P4M-SidM (Hammond et al., 2014; plasmid 51472; Addgene), mRFP-PH-PLCδ1, mRFP-PH-Akt, GFP-Sac2 (Nakatsu et al., 2015; gift from Pietro DeCamilli, Yale University, New Haven, CT), GFP-Rab27a, GFP-Rab3a, NPY-GFP, NPY-mCherry, VAMP2-pHluorin (all gifts from Sebastian Barg, Uppsala University, Uppsala, Sweden), Lamp1-RFP (Sherer et al., 2003; plasmid 1817; Addgene), GFP-Rab5, GFP-EEA1 (gifts from Pietro DeCamilli), CIBN-CAAX (Idevall-Hagren et al., 2012), GFP-CRY2-5ptase$_{OCRL}$ (Idevall-Hagren et al., 2012), GFP-CRY2-iSH2 (Idevall-Hagren et al., 2012), R-GECO1 (Zhao et al., 2011;

plasmid 32444; Addgene), GFP-Lact-C2 (Yeung et al., 2008; plasmid 22853; Addgene), EGFP-Rab4a (Rzomp et al., 2003; plasmid 49434; Addgene), GFP-Rab10 (Huckaba et al., 2011; plasmid 31737; Addgene), Granuphilin-GFP (Gálvez-Santisteban et al., 2012; plasmid 40032; Addgene), and GFP-2xFYVE$_{EEA1}$ (Petiot et al., 2003).

## Cloning

mCherry-Sac2 was generated by restriction enzyme digestion to release full-length mouse Sac2 from Sac2-GFP followed by ligation to mCherry-C1 (Clontech). Catalytically inactive GFP-Sac2(D460N) was generated by site-directed mutagenesis (Q5 site-directed mutagenesis; New England Biolabs) using the following primers: Sac2-DN forward (fwd), 5′-TAATTGCATGAA CTGCCTGGATC-3′; and Sac2-DN reverse (rev), 5′-ACTCGAAAA ATCCCTTCTTG-3′. GDP-locked forms of GFP-Rab3a (T36N) and GFP-Rab27a (T23N) were generated by site-directed mutagenesis using the following primers: Rab27a-DN-fwd, 5′-GTAGGG AAGAACAGTGTACTTTAG-3′; Rab27a-DN-rev, 5′-ACCAGAGTC TCCCAAAGC-3′; Rab3a-DN-fwd, 5′-GTGGGCAAAAACTCATTC CTT-3′; and Rab3a-DN-rev, 5′-ACTGCTGTTACCAATGAT-3′. shRNA-resistant mCherry-Sac2 was generated by site-directed mutagenesis using the following primers: Sac2-shres-fwd, 5′-GTTCGAAAATGTTCAAACCCTCACGGATG-3′; and Sac2-shres-rev, 5′-TTCATGCCTCGGCAGTGCTCATGGAAATC-3′. CIBN-GFP-Rab3a was generated by PCR amplification of rat Rab3a using the following primers with flanking Pvu1–BsrG1 restriction enzyme sites: rRab3A_Pvu1-fwd: 5′-TACGATCGATGGCCT CAGCCACAGACT-3′, rRab3A_BsrG1-rev: 5′-GTGTACATCAGC AGGCGCAATCCTGA-3′. The PCR product was subsequently ligated to C-terminus of CIBN-GFP (Idevall-Hagren et al., 2012). A version lacking GFP was also generated. GFP-CRY2-Sac2 was generated by PCR amplification of the 4′-phosphatase domain of mouse Sac2 (corresponding to amino acids 1–590) using the following primers: Pvu1-INPP5F-fwd: 5′-TACGATCGATGGAGC TCTTTCAGG-3′, Kpn1-INPP5F(1–590)-rev: 5′-GGGGTACCTCAC TCCTTATGCAAAGCT-3′. The PCR product was subsequently ligated to GFP-CRY2-5ptase$_{OCRL}$ by replacing 5ptase$_{OCRL}$.

## Cell culture and transfection

MIN6 cells (Miyazaki et al., 1990) were used between passages 22 and 35 and maintained in DMEM with 4.5 g/liter glucose, 2 mM L-glutamine, 100 U/ml Penicillin, 100 µg/ml Streptomycin, 50 µM 2-mercaptoethanol, and 15% fetal bovine serum (all from Life Technologies) in a humidified atmosphere at 37°C and 5% $CO_2$. The cell line was regularly checked for mycoplasma infections (MycoplasmaCheck; GATC Biotech). Transfections were performed on 25-mm poly-L-lysine–coated coverslips in 100 µl OptiMEM (Life Technologies) using 0.5 µl Lipofectamine 2000 (Life Technologies), 0.2–0.4 µg plasmid DNA, and 200,000 MIN6 cells. The reaction was terminated after 3–5 h, and imaging was performed 22–30 h after transfection.

## Stable and transient KD

Transient KD was performed using scrambled control siRNA and the following predesigned siRNAs (Ambion Life Technologies) targeting the indicated mouse gene sequence: Inpp5f_1: 5′-GCU

CUUAUAUCUCGAAGAATT-3′; and Inpp5f_2, 5′-GGAAUGCGG UAUAAACGAATT-3′. Pi4k2a and Pi4k2b silencing was performed using On-Target plus SmartPools (Dharmacon). Transfections were performed with Lipofectamine RNAiMAX (Life Technologies) according to the manufacturer's instructions with a final siRNA concentration of 10 nM. Stable KD MIN6 cell lines were established using the following shRNA (Mission ON-target plus shRNA; Sigma-Aldrich): 5′-CCGGCGAGGAATGAAGTTCGA GAATCTCGAGATTCTCGAACTTCATTCCTCGTTTTTG-3′ (shRNA3) and 5′-CCGGGCCTACTATGTGGCCTATTATCTCGAGATA ATAGGCCACATAGTAGGCTTTTTG-3′ (shRNA4). A shRNA with scrambled sequence was used as a control (plasmid 10879; 5′-CCG CAGGTATGCACGCGT-3′; Addgene). Cells with stable expression were selected in culture medium supplemented with 2 µg/ml Puromycin for 2 wk. Thereafter, cell lines were maintained in culture medium containing 1 µg/ml Puromycin. KD was routinely confirmed by quantitative RT-PCR.

## Quantitative RT-PCR

mRNA from control and Sac2 KD MIN6 cells and from mouse tissues were extracted using NucleoSpin RNAPlus kit (Macherey-Nagel). RT-PCR were performed using QuantiTect SYBR Green RT-PCR kit (Qiagen) with the following primers: Sac2-fwd, 5′-TAAGGAGAGCCAGAGAAGCCA-3′; Sac2-rev, 5′-CAGCAGCACTTC CACATCTCT-3′; PI4K2A-fwd, 5′-CCTGAGAACACGAACCGACA-3′; PI4K2A-rev, 5′-GTTGTCTATGGCAGCCACCT-3′; PI4K2B-fwd, 5′-TTCTGAAGCTGGTGCCTACC-3′; PI4K2B-rev, 5′-TGCCTCTTGATT TTGCACGG-3′; GAPDH-fwd, 5′-ACTCCACTCACGGCAAATTC-3′; GAPDH-rev, 5′-TCTCCATGGTGGTGAAGACA-3′. PCR reactions were performed using Light Cycler 2.0 (Roche), and results are expressed as ΔΔCt, normalized to the expression in islets from mouse tissues and to the expression in control samples (scramble shRNA or negative siRNA) in transient and stable KD cells.

## Immunoblotting

48 h after transfection with control or Sac2 siRNA, MIN6 cells were lysed on ice with RIPA buffer (50 mM Tris-HCl, pH 7.4, 1% NP-40, 0.5% Na-deoxycholate, 0.1% SDS, and 10% EDTA) supplemented with a protease inhibitor cocktail (Roche). Samples were cleared by centrifugation and protein concentration was measure by Bradford assay. Western blot was performed with antibodies against GFP (mouse monoclonal, clone 3E6, A-11120, 1:1,000; Thermo Fisher Scientific) and β-actin (mouse monoclonal, clone C4, sc-47778, 1:300; Santa Cruz Biotechnology).

## Fluorescence microscopy

All experiments, unless otherwise stated, were performed at 37°C in an experimental buffer containing 125 mM NaCl, 4.9 mM KCl, 1.2 mM $MgCl_2$, 1.3 mM $CaCl_2$, 25 mM Hepes, 3 mM D-Glucose, and 0.1% BSA (pH 7.40). For docking and exocytosis experiments, cells were kept in the experimental buffer supplemented with 20 mM D-Glucose and 100 µM diazoxide. Cells were preincubated in imaging buffer for 30 min before experiments, and continuously perfused with the same buffer at the stage of the microscope. Confocal microscopy was performed on a Nikon Eclipse TE-2000 equipped with a Yokogawa CSU-10 spinning disc confocal unit and a 100×/1.49-NA plan Apochromat

objective (Nikon) as previously described (Idevall-Hagren et al., 2013). Briefly, excitation light was provided by 491-nm and 561-nm DPSS lasers (Cobolt), and fluorescence was detected through a 530/50-nm interference filter (GFP) or a 597LP filter (mCherry) with a back-illuminated EMCCD camera (DU-888; Andor Technology) controlled by MetaFluor software (Molecular Devices). TIRF microscopy was performed on two different setups. A custom-built prism-type TIRF microscope equipped a 16×/0.8-NA water-immersion objective (Nikon) was used for low-magnification imaging of cell populations as previously described (Idevall-Hagren et al., 2013). It is built around an E600FN upright microscope (Nikon) contained in an acrylic glass box thermostated at 37°C by an air stream incubator. DPSS lasers (Cobolt) provided 491-nm and 561-nm light for excitation of GFP and mCherry. The laser beams were merged with dichroic mirrors (Chroma Technology), homogenized, and expanded by a rotating Light Shaping Diffuser (Physical Optics) before being refocused through a modified quartz dove prism (Axicon) with a 70° angle to achieve total internal reflection. Laser lines were selected with interference filters (Semrock) in a motorized filter wheel equipped with a shutter (Sutter Instruments) blocking the beam between image captures. Fluorescence from the cells was detected at 530/50 nm for GFP (Semrock interference filters) or 597LP for mCherry (Melles Griot glass filter) with a CCD camera (ORCA-AG) under MetaFluor software control. For colocalization studies, an objective-type microscope (Eclipse TiE; Nikon) equipped with 60×/1.45-NA or 100×/1.49-NA plan Apochromat TIRF objectives (Nikon) was used instead, as previously described (Dyachok et al., 2008). Visualization of single granule docking and exocytosis was performed on a custom-built TIRF microscope based on an AxioObserver D1 microscope and a 100×/1.45-NA objective from Carl Zeiss. Excitation was from two DPSS lasers at 491 and 561 nm from Cobolt, controlled with an acousto-optical tunable filter (AA-Opto) and using dichroic Di01-R488/561 from Semrock. The emission light was separated onto the two halves of a 16-bit EMCCD camera from Roper (Cascade 512B) with constant gain setting throughout. The setup was equipped with an image splitter from Photometrics (DV2) and ET525/50-nm and 600/50-nm emission filters from Chroma. Exocytosis was evoked with 75 mM $K^+$ (equimolarly replacing $Na^+$) applied by computer-timed local pressure ejection through a pulled glass capillary. Scaling was 100 nm per pixel.

### Optogenetic manipulation of phosphoinositides
In microscopy experiments, light-induced dimerization of CRY2 and CIBN was achieved by the same 491-nm DPSS laser that was used for imaging (see above). In some experiments, dimerization was instead achieved by epifluorescence illumination using a collimated 470-nm LED diode (Mightex Systems) as previously described (Xie et al., 2016). For measurements of GH release combined with optogenetic manipulations, MIN6 cells expressing GH were cultured in 96-well plates and exposed to blue light using a wireless LED array, as recently described (Qi, 2018). All optogenetic manipulations were performed at 37°C.

### Insulin secretion measurements
MIN6 cells were seeded in 6-well (insulin) or 96-well (GH) plates and allowed to settle for 24–48 h. Subsequently, cells were washed three times with PBS and preincubated in experimental buffer (125 mM NaCl, 4.9 mM KCl, 1.2 mM $MgCl_2$, 1.3 mM $CaCl_2$, 25 mM Hepes, and 0.1% BSA) supplemented with 3 mM D-glucose for 30 min at 37°C. This was followed by 30-min incubation in experimental buffer containing 3 mM or 20 mM D-glucose, and the supernatant from each well was collected and hormone concentration was measured using AlphaLISA insulin or GH kits (Perkin-Elmer). The cells were subsequently trypsinized and collected in acidic ethanol (75% EtOH 95–99% and 15% HCl 37%), sonicated 2 × 10 s on ice, and diluted 1:10 in Tris, pH 8.0, for total hormone content measurements. The human islet stimulation index was routinely measured and provided by the islet isolation center. It represents the ratio of peak insulin secretion in 16.7 mM glucose and basal insulin secretion in 5.6 mM glucose, measured by superfusion of islet batches (Goto et al., 2004).

### RNA-sequencing and gene expression analysis
Human islets were obtained from the Nordic Network for Clinical Islet Transplantation Uppsala under full ethical clearance (Uppsala Regional Ethics Board 2006/348) and with the donor families' written informed consent. RNA-sequencing and gene expression analysis was performed as described previously (Fadista et al., 2014). Briefly, the quality of the isolated RNA was assessed by 2100 Bioanalyzer (Agilent Technologies) or 2200 Tapestation (Agilent Technologies), and quantity was measured on a NanoDrop 1000 (NanoDrop Technologies) or a Qubit 2.0 Fluorometer (Life Technologies). 1 µg total RNA was used for sample preparation for sequencing with a TruSeq RNA sample preparation kit (Illumina). Size selection was done using Agencourt AMPure XP beads (Beckman Coulter) aiming at a fragment size >300 bp. The resulting libraries were quality controlled on a 2200 Tapestation (Agilent Technologies) before combining six samples into one pool for sequencing on one lane on a flow cell sequenced on a HiSeq 2000 (Illumina). The raw RNA-sequencing data were base called and demultiplexed using CASAVA 1.8.2 (Illumina) before alignment to hg19 with STAR. To count the number of reads aligned to specific transcripts, featureCounts (http://bioinf.wehi.edu.au/featureCounts/) was used. Raw data were normalized using trimmed mean of M-values and transformed using voom (limma R-package) before linear modeling. All expression values are expressed as log2 counts per million after normalization and transformation using voom. The expression data were previously published and are publicly accessible at Gene Expression Omnibus (GEO accession numbers GSE50398 and GSE108072).

### Image analysis
TIRF microscopy images were analyzed offline using ImageJ. Briefly, for analysis of VAMP2-pHluorin and Fluo4 experiments (Fig. 4), cell footprints were manually identified and regions of interest covering the edges of the adherent cells were drawn. Intensity changes within these regions during the experiments were measured and exported to Excel. All data points were background corrected, followed by normalization to the prestimulatory level ($F/F_0$). For colocalization analysis, organelle marker–positive structures in cells coexpressing fluorescence-tagged Sac2

were identified and 30 × 30-pixel squares were drawn with the structure in the center and saved as separate images. For each cell, a minimum of 15 structures were randomly selected and an average projection image was generated. The line profiles from these images were measured and normalized to the minimal background fluorescence value ($min$). To estimate the enrichment of Sac2 and other proteins at the granule site, we first determined the width of the granule ($N_{gr}$). Next, we integrated the area under the curve for the line profiles of Sac2 (or other proteins; $AUC_{gr}$) and the whole range ($AUC_{All}$). Enrichment was calculated as $[AUC_{gr} - N_{gr} \times min]/[AUC_{All} - N_{All} \times min]$. If there is no enrichment of Sac2 at the granule site, this value should approach $N_{gr}/N_{All}$. Exocytosis events were identified manually based on the characteristic rapid loss of the granule marker fluorescence within one or two frames and analyzed as described previously (Gandasi et al., 2015). Granule density was calculated using a script with the built-in "find maxima" function in ImageJ (http://rsbweb.nih.gov/ij) for spot detection (Gandasi and Barg, 2014). Docking or undocking events were found manually as granules approaching the TIRF field axially to become laterally confined for ≥2 s. Docking in Fig. 5 is defined as granules that remain confined for ≥25 s. Granules that remained for <25 s at the plasma membrane were referred to as visitors.

### Statistical analysis

Statistical analysis was performed using two-way ANOVA followed by Tukey's post hoc test (for multiple comparisons) or two-tailed (paired or unpaired) Student's $t$ test with the significance level set to $P < 0.05$.

### Online supplemental material

Fig. S1 shows the lipid specificity of the optogenetic tools used in this study. Fig. S2 shows examples of the colocalization between Sac2 and markers of the exocytic and endocytic pathways. Fig. S3 shows examples of the cellular distribution of Sac2 after pharmacological inhibition or siRNA-mediated KD of PI4 kinases. Fig. S4 shows the efficiency of Sac2 KD in MIN6 cells. Fig. S5 shows the distribution of Rab27, Rab3, and granuphilin after Sac2 KD.

## Acknowledgments

We are grateful to Antje Thonig and Petra Franzen for expert technical assistance and Erik Gylfe and Anders Tengholm for comments on the manuscript.

O. Idevall-Hagren was supported by the Swedish Research Council (MH-2015-03087), the Novo Nordisk Foundation (NNF15OC0016100 and NNF19OC0055275), Göran Gustafssons stiftelse, Malin och Lennart Philipsons stiftelse, Magnus Bergwalls stiftelse, Åke Wibergs stiftelse, Exodiab, and STINT (Stiftelsen för internationalisering av högre utbildning). N.R. Gandasi was supported by the Novo Nordisk Foundation. S. Sugahara was supported by The Graduate Program for Leaders in Life Innovation, the University of Tokyo (part of the Program for Leading Graduate Schools, MEXT, Japan).

The authors declare no competing financial interests.

Author contributions: Conceptualization, O. Idevall-Hagren; Investigation, P.M. Nguyen, N.R. Gandasi, S. Sugahara, B. Xie, and O. Idevall-Hagren; Formal Analysis, P.M. Nguyen, N.R. Gandasi, and O. Idevall-Hagren; Providing Key Reagents, Y. Xu; Writing – Original Draft, O. Idevall-Hagren; Writing – Review & Editing, P.M. Nguyen, N.R. Gandasi, and O. Idevall-Hagren; Supervision, O. Idevall-Hagren; Funding Acquisition, O. Idevall-Hagren.

Submitted: 22 March 2019

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
