## [Reviewer comments · The Journal of Cell Biology]

The PI(4)P phosphatase Sac2 controls insulin granule docking and release

Phuoc Nguyen, Nikhil Gandasi, Beichen Xie, Sari Sugahara, Yingke Xu, and Olof Idevall-Hagren

Corresponding Author(s): Olof Idevall-Hagren, Uppsala University, Dept. Medical Cell Biology

Review Timeline:

Submission Date:	2019-03-22
Editorial Decision:	2019-04-16
Revision Received:	2019-06-20
Editorial Decision:	2019-07-11
Revision Received:	2019-08-06

Monitoring Editor: Vivek Malhotra

Scientific Editor: Melina Casadio

Transaction Report:

DOI: <https://doi.org/10.1083/jcb.201903121>

April 16, 2019

Re: JCB manuscript #201903121

Dr. Olof Idevall-Hagren
Uppsala University, Dept. Medical Cell Biology
Husargatan 3
BMC
Uppsala 75123
Sweden

Dear Dr. Idevall-Hagren,

Thank you for submitting your manuscript entitled "The PI(4)P phosphatase Sac2 controls insulin granule docking and release". The manuscript was assessed by expert reviewers, whose comments are appended to this letter. We invite you to submit a revision if you can address the reviewers' key concerns, as outlined here.

You will see that the reviewers all found the link between Sac2 and insulin granule docking and release quite interesting. However, they shared numerous experimental and technical issues that require your attention, experimentally or textually, to strengthen this intriguing new conclusion. To summarize their issues, Rev#1 is not satisfied with the approaches used to show that PI4-kinase is not needed for Sac2 targeting to the granules (point #1) and recommends more refined genetic analyses and another inhibitor to tease apart which isoforms may or may not be involved. Rev#1 would like you to actually test the model and look at whether Sac2 may be involved in a potential conversion from PI(4)P to PI(3)P (if occurring) (point #2) and test the link between Sac2 and granophilin targeting to the insulin granules (point #3). Rev#3 echoes this concern by noting that the "primary weakness of the paper is the lack of mechanistic insight into how PI4P is regulating granule docking" and Rev#3 seems interested in testing whether lack of Sac2/ excess PI(4)P accumulation in the granules leads to an excess cholesterol loading driven by OSBP-mediated PI4P/cholesterol exchange (point #1). Ref#1 suggests a more thorough analysis of Rabs potentially involved in insulin granule trafficking (point#6, beyond Rab3/27a that are already tested) and Rev#2 adds that other factors should be looked at too, like SNAP25 and Syntaxin-1 (last specific comment). Rev#1 asks if Sac2 could be involved in PI(4)P/PS exchange by ORP family proteins on insulin granules (Rev#1 point #5). Rev#2 asks for detection of endogenous Sac2 on insulin granules (and we appreciate this has been a challenge for other teams studying Sac2 due to the lack of working antibodies, please see our recommendation below on the points on which to focus instead in revision) and to detect PI(4)P on purified granules (point #7). Rev#2 suggests showing that the docking phenotype is due to Sac2's enzymatic activity and not a structural/other role (specific comment #9). Rev#2 recommends edits to the text to enhance readability and flow and suggests adding more info about the methods (#1). Rev#2 also asks for data that does not seem to be shown but is described in the paper - showing that the same amounts of granules approach the PM in Sac2 WT and KD cells (point #2). The ref points out that Sac2 may also play a similar role in other compartments (#3) and asks if constitutive secretion also requires prior depletion of PI(4)P from granules (#5). Rev#2 comments on the low numbers of cells used in some experiments (#6) and has other suggestions for improvement in terms of clarity and data presentation (specific comments). Rev#3 has suggestions for clarifications (#2-3-4) and suggests discussing the physiological

implications of these results in vivo (last point).

We agree with the reviewers that the main claims in the study should be bolstered for publication, and from the revs' comments, our editorial assessment is that the revision should focus on strengthening the core conclusions in the study and addressing the following points:

1. Please knock down PI4K2A and PI4K2B and examine Sac2 localization. We recommend employing PIK93, a specific inhibitor of type III PI4K beta, to further examine PI4K requirements.
2. We agree with the revs about testing whether recruitment of CRY2-Sac1 results in rapid GFP-FYVE (PI3P reporter) targeting to insulin granules.
3. Does depletion of Sac2 inhibit PI3P synthesis on the granules and block granuphilin targeting?
4. Does depletion of Sac2 impair targeting of the PS reporter GFP-LactC2 to insulin granules?
5. Please quantitate the number of granules approaching the surface (point #2 of 2nd reviewer).
6. Better images and clarifications of figures suggested by Reviewer #2 (points #4 and #6) need to be provided.
7. Reviewer #2 has listed a number of concerns regarding data presentation (specific points 1-10 -- including the rev's suggestion to test for the requirement of phosphatase function) that should be addressed.
8. Reviewer #3 has requested changes in the text and figures that should be addressed. The mouse KO of Sac2 is certainly a valuable tool for further analysis, but not necessary for this paper (but please do respond to the reviewer's point in the text). Please consider all the reviewers' suggestions for text edits and clarifications/discussion carefully.

Please let us know if you would like to discuss the revisions further or anticipate any issues addressing the points highlighted in this letter. While you are revising your manuscript, please also attend to the following editorial points to help expedite the publication of your manuscript. Please direct any editorial questions to the journal office.

GENERAL GUIDELINES:

Text limits: Character count for an Article is < 40,000, not including spaces. Count includes title page, abstract, introduction, results, discussion, acknowledgments, and figure legends. Count does not include materials and methods, references, tables, or supplemental legends.

Figures: Articles may have up to 10 main text figures. Figures must be prepared according to the policies outlined in our Instructions to Authors, under Data Presentation, <http://jcb.rupress.org/site/misc/ifora.xhtml>. All figures in accepted manuscripts will be screened prior to publication.

Supplemental information: There are strict limits on the allowable amount of supplemental data. Articles may have up to 5 supplemental figures. Up to 10 supplemental videos or flash animations are allowed. A summary of all supplemental material should appear at the end of the Materials and methods section.

The typical timeframe for revisions is three months; if submitted within this timeframe, novelty will

not be reassessed at the final decision. Please note that papers are generally considered through only one revision cycle, so any revised manuscript will likely be either accepted or rejected.

We hope that the comments below will prove constructive as your work progresses. We would be happy to discuss them further once you've had a chance to consider the points raised in this letter. Thank you for this interesting contribution to the Journal of Cell Biology. You can contact us at the journal office with any questions, cellbio@rockefeller.edu or call (212) 327-8588.

Sincerely,

Vivek Malhotra, PhD
Monitoring Editor, Journal of Cell Biology

Melina Casadio, PhD
Senior Scientific Editor, Journal of Cell Biology

Reviewer #1 (Comments to the Authors (Required)):

In this study, Nguyen et al. reveal a previously unknown role for the Sac2 phosphoinositide phosphatase in insulin granule secretion. They find that Sac2 localizes to insulin granules and that loss of Sac2 results in impaired insulin granule docking at the plasma membrane. These are interesting observations with potentially important physiological implications, as Sac2 expression is decreased in islet cells isolated from type 2 diabetics.

Based on their findings, the authors conclude that Sac2 mediates a phosphoinositide lipid conversion step from PI4P to PI3P prior to insulin granule docking. This step recruits PI3P-binding proteins necessary for insulin granule docking. The experimental data are of high quality and lend support to the overall model. However, a few issues and questions remain unresolved. In particular, a few additional experiments are needed to further clarify the role for Sac2 function in insulin granule docking.

Major comments

1. The authors take advantage of a CRY2-Sac1 fusion to acutely deplete the phosphoinositide species PI4P from insulin granules. This sophisticated approach showed that PI4P is needed for Sac2 localization to insulin granules. However, the authors conclude that PI4P synthesis on insulin granules per se is not required for Sac2 targeting. This is based on crude experiments using the inhibitors wortmanin and PAO. However, only type III PI 4-kinase alpha is highly sensitive to these compounds, and type II PI 4-kinase isoforms are not significantly inhibited. It is therefore unclear whether type II PI4Ks or type III PI4K beta generates the PI4P present on insulin granules. As type II PI4Ks have been implicated in regulated exocytosis in numerous studies, the authors should knock down PI4K2A and PI4K2B and examine Sac2 localization. The authors may also want to test PIK93, a specific inhibitor of type III PI4K beta, to further examine PI4K requirements.

2, The authors propose an interesting model whereby Sac2-mediated PI4P hydrolysis is a

prerequisite for PI3P synthesis on insulin granules. This would fit with previous studies showing that PI3P and the FYVE domain-containing protein granuphilin are required for insulin granule docking. However, the authors have not directly demonstrated in this study whether a phosphoinositide conversion from PI4P to PI3P actually occurs. If so, is Sac2 involved? The authors should examine whether recruitment of CRY2-Sac1 results in rapid GFP-FYVE (PI3P reporter) targeting to insulin granules. Also, does depletion of Sac2 impair PI3P synthesis on insulin granules?

3. Likewise, according to the authors' model (as proposed in the Discussion), Sac2 may control granuphilin targeting to insulin granules. The authors should examine whether recruitment of CRY2-Sac1 results in rapid granuphilin targeting to insulin granules. Likewise, does depletion of Sac2 impair granuphilin targeting to insulin granules?

4. Alternative roles for Sac2 function during insulin granule docking may exist. Does Sac2-mediated PI4P hydrolysis promote a Rab GTPase cascade as proposed for the yeast secretory pathway? Although Sac2 does not appear to be involved in Rab3 localization, the authors may want to examine the late-acting Rab10 as well as Rab27. For example, is Rab10 present on insulin granules? If so, does recruitment of CRY2-Sac1 result in rapid Rab10 targeting to insulin granules? Does depletion of Sac2 impair Rab10 targeting?

5. As mentioned in the Discussion, phosphatidylserine is needed for efficient insulin granule docking. Might Sac2 promote PI4P/PS exchange by ORP family members on insulin granules? Does depletion of Sac2 impair targeting of the PS reporter GFP-LactC2 to insulin granules?

Minor comments

1. The level of PI4P on insulin granules is grossly exaggerated in the Discussion. The MacDonald et al 2015 study measured phosphatidylinositol (PtdIns), not phosphoinositides. PtdIns (not PI4P!) makes up around 20% of the lipids on insulin granules. PI4P is usually around 2% of PtdIns and thus PI4P is approximately only 0.4% of lipids on insulin granules.

2. Sac1 and Sac2 are not the only 4'-PI phosphatases in mammalian cells as stated in the Introduction. For example, the synaptojanin proteins possess a Sac1 domain.

Reviewer #2 (Comments to the Authors (Required)):

Summary

This study presents evidence for the participation of Sac2, a predominantly neuronal PI(4)P phosphatase, in insulin secretion from pancreatic β -cells (MIN6 cell line). Upon overexpression Sac2 colocalized with NPY, Rab3 and Rab27, known to be on insulin granules. Using optogenetics and knockdown experiments it is proposed that localization depended on granular PI4P and Rab3A. Sac2 knockdown reduced glucose-stimulated insulin secretion (GSIS) from cultured β -cells, studied using phluorin-based fusion assays and surrogate exocytosis of growth hormone. The authors propose that the reduced exocytosis is due to a defect in vesicle docking at the plasma membrane, brought about by prior depletion of PI from granules (4). This conclusion is based on the exclusionary observations/statements, that the number of granules approaching the membrane is not changed by Sac2KD, but exocytosis is reduced, and that the parameters of fusion are also not affected. The step in between is by exclusion deemed to be granule docking.

Major Comments:

1) Identifying that loss of PI(4) is a requirement for subsequent granule docking and that this is mediated by Sac2 binding to the maturing granule, is an important finding. However, this message is obscured by the dense writing of the manuscript. This applies to the description of the experimental design and discussion of the results. While the rationale is supported for each experiment, the methodologies and data are communicated in unclear ways that make the findings difficult to understand and interpret. More detail regarding methodologies would greatly enhance the readability of the manuscript (see examples under Specific comment 1).

2) The statement that "a similar number of granules approached the plasma membrane in both control and Sac2D cells" is crucial for the interpretation of the entire paper, as it is a basis for concluding that the reduction in docking is the major consequence of Sac2D depletion. The experiment measuring the number of granules approaching must be presented.

3) The requirement of Sac2 activity may not be exclusive to insulin granules. Example: PMID: 2803504 Role of Sac2 in decreasing PI(4)P levels as the phagosome seals in macrophages.

4) Single puncta (granules?) are shown in various cases (Fig 1E, Fig 2 I-K, Fig 3A). It is not possible to assess whether this is reflective of the population that exists within a cell. It would be preferable to include whole-cell images with multiple magnified ROI's containing individual puncta to illustrate these points, in order to more firmly support the derived conclusions. In Fig 1E, the point is to show that Rab3 is peripheral and NPY is luminal. This is not convincingly conveyed. In addition, Rab3A is not only on insulin granules, this must be clarified, and the data interpreted accordingly.

5) Does constitutive secretion, mentioned in the Introduction, also require prior depletion of PI(4)P from granules?

6) Very few of cells were scrutinized in most instances and this requires clarification: Examples: Figure 1F: < 10 cells; Figure 2C,D: 8-12 cells; Figure 2F: 7-11 cells; Figure 2H: 6-7 cells; Figure J: 6 cells; Figure 2K,L: 11 cells

7) It would be important to detect the localization of the endogenous Sac2 on insulin granules (by colocalization with insulin or growth hormone) as well as to detect PI(4) on purified granules. Moreover, some discussion is required as to whether the endogenous levels of Sac2 are limiting, given that overexpressing Sac2-GFP enhanced granule fusion.
Specific Comments:

1) Examples of lack of clarity: Supplemental Figure 1A-C: the authors validate that each of the phosphoinositide-modifying constructs used had impact on plasma membrane phosphoinositides. There is a key detail missing from this figure legend and the in-text explanation. In contrast to the results Figure 1A-C where the constructs were targeted to the granular membrane, Supplemental Figure 1A-C instead uses a plasma membrane targeted system that is NOT described (in text or in the figure legend) to verify that each of the constructs was with effect on the plasma membrane. This is distinct from the immediately subsequent Supplemental Figure D-G, where the authors again use the granular membrane-targeted constructs to show that they are without effect on plasma membrane phosphoinositides. Given that these two figures are showing seemingly opposite results (A-C having an effect on PM phosphoinositides, and D-G without effect on PM phosphoinositides), it is crucial to clearly specify these inferred details for clarity and understandability.

- 2) Figure 1C: Overexpression of a granule-targeted Sac1 potentiated glucose-stimulated growth hormone secretion. It is essential to show that expression of a granule-targeted Sac2 phosphatase has the same effect.
- 3) Figure 1D-F: The text states "we expressed GFP-tagged Sac2 in MIN6 B-cells and imaged the cells by both TIRF and confocal microscopy", but these data are not shown. Instead, the authors present data of a phosphatase-dead DN Sac2 mutant (Fig 1D) and study the localization of this mutant with markers of insulin granules and early and late endosomes (1E,F). Clarification regarding the choice of the DN Sac2 mutant (over the WT) would be helpful. Examples: Figure 1E: images of single granules show co-localization of Sac2 with: Rab27a, Rab3a, and NPY. However, the authors report in Figure 1F that Sac2 co-localizes with additional markers (not shown in 1E): EEA1, Rab5, Lamp1. Given that images for Sac2 co-localization with either early endosome marker were not shown and that Sac2 shows the highest enrichment in this compartment, images are relevant. Figure 1E: y axis 'Fold enrichment' - it is unclear exactly which parameters were used to measure Fold enrichment (not clarified in text or figure caption).
- 4) Figure 2F: It is concluded that PI4P synthesis is not required for Sac2 localization to the insulin granule since inhibition of PI4K did not alter Sac2 enrichment on the granular membrane. However, the data appear to trend towards an involvement of PI4K/PI4P synthesis, given that PAO treatment trends towards decreasing Sac2 enrichment in insulin granules. This in conjunction with only 11 cells tested, make it premature to conclude that there is no role for PI4K in Sac2 targeting to insulin granules.
- 5) Figure 2G&H: Inconsistent treatment group notation. CRY2-rptase and opto-4'-ptase are two different notations used to refer to the same construct in a single figure.
- 6) Figure 3E: Inconsistent treatment group notation. Knockdown cells are referred to as "siRNA #2", whereas in the remaining panels of the figure, the notation Sac2 KD is used.
- 7) Figure 5A: This illustration is depicting 'the principle for local, acute stimulation of exocytosis from single cells', however more detail is required in text.
- 8) Figure 5C states that Sac2 KD cells have reduced vesicle density at the plasma membrane. Referencing Figure 5E the text states that "a similar number of granules approached the plasma membrane in both control and Sac2D cells", but the data are not shown. Given the clear reduction in vesicle density at the plasma membrane in Sac2 KD cells (Fig 5C), the data in Figure 5E should be revised to report the absolute number of vesicles approaching the plasma membrane (currently reported as %).
- 9) Figure 5E: Does expression of WT-Sac2 recover the defects in granule docking observed in Sac2 KD? Similar data has been shown for Rab3A and Rab27A (both of which are reduced in T2D islets). The authors should consider experiments to prove that the impairment in docking is specifically due to the loss of Sac2 phosphatase activity, and not an auxiliary unknown action of Sac2 that could also contribute to vesicle docking independent of PI(4)P levels. Overexpression of granule-targeted Sac1 (used in Figure 1C) could be used in Sac2 KD cells to restore phosphatase activity against PI(4)P and determine whether or not this reverses impairments in docking could be identified.
- 10) Supplemental Figure 4: It is shown that docking proteins Rab3a and Rab27a are not lost from granules in Sac2 KD cells, but other factors must be considered, such as SNAP25 and Syntaxin-1.

Reviewer #3 (Comments to the Authors (Required)):

Defining mechanisms for how membranes are remodeled within the secretory pathway and how this links to protein trafficking events is an important endeavor. Here, the authors have discovered a role for Sac2 in regulating PI4P levels on insulin granules and provide convincing evidence for the functional significance of this regulation in cultured cells. Some very nice tools for optogenetic erasure of PI4P from insulin granules and measuring exocytosis using pHluorin fusions were developed and used for this study. Live cell imaging data also implicated granule docking at the plasma membrane as being the primary step requiring PI4P degradation by Sac2. The primary conclusions are supported well by the data. The primary weakness of the paper is the lack of mechanistic insight into how PI4P is regulating granule docking. It can be very challenging to identify the relevant effectors of a phosphoinositide, but reasonable candidates could be assessed. In this case, OSBP and cholesterol have also been linked to insulin granule maturation (PMID:29540530). One predicted consequence of Sac2 depletion and excess PI4P accumulation in the granules would be excess cholesterol loading driven by OSBP-mediated PI4P/cholesterol exchange (PMID:28978670).

1. Would it be possible for the authors to test whether OSBP is the relevant PI4P effector that is perturbing granule docking in Sac2 deficient cells? The citations provided above should at least be introduced and discussed in the context of the current work.
2. There are a number of typos in the manuscript and descriptions of data in the text were often insufficient to accurately communicate how the experiments were done. For example, OCRL and iSH2 in Fig 1C should be better defined in the results section.
3. In Figure 2E and other comparable graph the Y axis should be better defined. F/F0 would not generate a negative value. Isn't this F0-F?
4. Supplemental Fig 1 inappropriately truncates the Y-axis to exaggerate effects. The GRP1 designation in B is different from what is in the legend, which uses PH-Akt.
5. Lastly, there is a mouse knockout of Sac2 and I don't see any indication that there is hypoinsulinemia in the mice, although it could be that no one has looked. I think it would be beyond the scope of the current work, but future studies should address the physiological significance of these observations.

Reviewer #1 (Comments to the Authors (Required)):

In this study, Nguyen et al. reveal a previously unknown role for the Sac2 phosphoinositide phosphatase in insulin granule secretion. They find that Sac2 localizes to insulin granules and that loss of Sac2 results in impaired insulin granule docking at the plasma membrane. These are interesting observations with potentially important physiological implications, as Sac2 expression is decreased in islet cells isolated from type 2 diabetics.

Based on their findings, the authors conclude that Sac2 mediates a phosphoinositide lipid conversion step from PI4P to PI3P prior to insulin granule docking. This step recruits PI3P-binding proteins necessary for insulin granule docking. The experimental data are of high quality and lend support to the overall model. However, a few issues and questions remain unresolved. In particular, a few additional experiments are needed to further clarify the role for Sac2 function in insulin granule docking.

Major comments

1. The authors take advantage of a CRY2-Sac1 fusion to acutely deplete the phosphoinositide species PI4P from insulin granules. This sophisticated approach showed that PI4P is needed for Sac2 localization to insulin granules. However, the authors conclude that PI4P synthesis on insulin granules per se is not required for Sac2 targeting. This is based on crude experiments using the inhibitors wortmanin and PAO. However, only type III PI 4-kinase alpha is highly sensitive to these compounds, and type II PI 4-kinase isoforms are not significantly inhibited. It is therefore unclear whether type II PI4Ks or type III PI4K beta generates the PI4P present on insulin granules. As type II PI4Ks have been implicated in regulated exocytosis in numerous studies, the authors should knock down PI4K2A and PI4K2B and examine Sac2 localization. The authors may also want to test PIK93, a specific inhibitor of type III PI4K beta, to further examine PI4K requirements.

Reply: We have now performed the suggested experiments. As also suggested by reviewer 2 (specific comment #4), we have repeated the experiments with wortmannin and PAO to see if the previously presented trend towards reduced granule association would reach significance if more cells were examined. We now report that PAO treatment results in partial loss of Sac2 from insulin granules, whereas wortmannin and PIK93 were without major effect. siRNA-mediated knockdown of PI4K2A/B (80% KD efficiency) was also without effect on Sac2 association with insulin granules. We have also repeated the experiments where a 4'-phosphatase is recruited to the granule surface, and now show in a more quantitative way its impact on Sac2 granule binding. These new results slightly alters the conclusions drawn in the paper, where we now state that Sac2 binding to insulin granules depend on PI(4)P that at least in part derives from the activity of a granule-localized PI4 kinase. The results are presented in new **Figure 2F-J** and **Supplementary figure 3**.

2, The authors propose an interesting model whereby Sac2-mediated PI4P hydrolysis is a prerequisite for PI3P synthesis on insulin granules. This would fit with previous studies showing that PI3P and the FYVE domain-containing protein granuphilin are required for insulin granule docking. However, the authors have not directly demonstrated in this study whether a phosphoinositide conversion from PI4P to PI3P actually occurs. If so, is Sac2 involved? The authors should examine whether recruitment of CRY2-Sac1 results in rapid

GFP-FYVE (PI3P reporter) targeting to insulin granules. Also, does depletion of Sac2 impair PI3P synthesis on insulin granules?

Reply: *Different to what has been shown in e.g. PC-12 cells, we do not observe any colocalization between a PI(3)P biosensor (GFP-2xFYVE^{EEA1}) and insulin granules either in control cells or in Sac2 KD cells. Acute recruitment of a 4'phosphatase to the granule surface did not result in any GFP-2xFYVE accumulation on the granule (New Figure 6A-D). We therefore do not believe that PI(3)P synthesis occurs downstream of Sac2 activity on the surface of insulin granules. We have revised the discussion accordingly.*

3. Likewise, according to the authors' model (as proposed in the Discussion), Sac2 may control granophilin targeting to insulin granules. The authors should examine whether recruitment of CRY2-Sac1 results in rapid granophilin targeting to insulin granules. Likewise, does depletion of Sac2 impair granophilin targeting to insulin granules?

Reply: *We readily detect GFP-granophilin on the surface of insulin granules in both control and Sac2 knockdown cells. The binding to insulin granules is not affected by reduced Sac2 levels (Supplementary figure 5D,E). Given that we were unable to detect any PI(3)P on the surface of insulin granules in either control or Sac2 knockdown cells, and that acute PI(4)P dephosphorylation on the granules failed to produce PI(3)P (see previous point), we did not assess the potential impact acute PI(4)P dephosphorylation on the granule surface had on granophilin binding.*

4. Alternative roles for Sac2 function during insulin granule docking may exist. Does Sac2-mediated PI4P hydrolysis promote a Rab GTPase cascade as proposed for the yeast secretory pathway? Although Sac2 does not appear to be involved in Rab3 localization, the authors may want to examine the late-acting Rab10 as well as Rab27. For example, is Rab10 present on insulin granules? If so, does recruitment of CRY2-Sac1 result in rapid Rab10 targeting to insulin granules? Does depletion of Sac2 impair Rab10 targeting?

Reply: *We have now examined the distribution of Rab10 and Rab27a in control and Sac2 KD cells. Rab27a, as already presented in the original version of the manuscript, is strongly enriched on insulin granules (and also colocalize with Sac2 on these structures) but this association is unaffected by Sac2 KD (Suppl. Fig. 5A,C). GFP-Rab10 localizes to vesicular structures, but there is no overlap between these and a marker of insulin granules (Suppl. Fig. 2D).*

5. As mentioned in the Discussion, phosphatidylserine is needed for efficient insulin granule docking. Might Sac2 promote PI4P/PS exchange by ORP family members on insulin granules? Does depletion of Sac2 impair targeting of the PS reporter GFP-LactC2 to insulin granules?

Reply: *This is a very interesting hypothesis, in particular since ORP5/8 has been shown to localize to more than one type of membrane contact site. We have now extensively tested this hypothesis. The PS sensor GFP-Lact-C2 localizes to insulin granules, both those that are docked and those that are located at a distance from the plasma membrane. As granules undergo exocytosis, there is a slow disappearance of GFP-LactC2 fluorescence from the exocytic site, consistent with lateral diffusion of PS in the plasma membrane following granule fusion. The GFP-Lact-C2 distribution and relative accumulation on insulin granules*

was not different in cells with reduced *Sac2* expression. These results are presented in new **Figure 6E-J**.

Minor comments

1. The level of PI4P on insulin granules is grossly exaggerated in the Discussion. The MacDonald et al 2015 study measured phosphatidylinositol (PtdIns), not phosphoinositides. PtdIns (not PI4P!) makes up around 20% of the lipids on insulin granules. PI4P is usually around 2% of PtdIns and thus PI4P is approximately only 0.4% of lipids on insulin granules.

Reply: We thank the reviewer for pointing this out. We have now made changes in the text accordingly:

“Insulin granules have a very high phosphatidylinositol content, but the relative abundance of its phosphorylated derivatives is not known (MacDonald et al., 2015).”

2. Sac1 and Sac2 are not the only 4'-PI phosphatases in mammalian cells as stated in the Introduction. For example, the synaptojanin proteins possess a Sac1 domain.

Reply: We thank the reviewer for pointing this out. We have now made changes in the text accordingly:

“PI(4)P dephosphorylation in mammalian cells is catalyzed by numerous PI(4)P phosphatases (Foti et al., 2001; Guo et al., 1999; Hsu et al., 2015; Nakatsu et al., 2015; Rohde et al., 2003).”

Reviewer #2 (Comments to the Authors (Required)):

Summary

This study presents evidence for the participation of Sac2, a predominantly neuronal PI(4)P phosphatase, in insulin secretion from pancreatic β -cells (MIN6 cell line). Upon overexpression Sac2 colocalized with NPY, Rab3 and Rab27, known to be on insulin granules. Using optogenetics and knockdown experiments it is proposed that localization depended on granular PI4P and Rab3A. Sacs knockdown reduced glucose-stimulated insulin secretion (GSIS) from cultured β -cells, studied using phluorin-based fusion assays and surrogate exocytosis of growth hormone. The authors propose that the reduced exocytosis is due to a defect in vesicle docking at the plasma membrane, brought about by prior depletion of PI from granules (4). This conclusion is based on the exclusionary observations/statements, that the number of granules approaching the membrane is not changed by Sac2KD, but exocytosis is reduced, and that the parameters of fusion are also not affected. The step in between is by exclusion deemed to be granule docking.

Major Comments:

1) Identifying that loss of PI(4) is a requirement for subsequent granule docking and that this is mediated by Sac2 binding to the maturing granule, is an important finding. However, this message is obscured by the dense writing of the manuscript. This applies to the description of the experimental design and discussion of the results. While the rationale is supported for each experiment, the methodologies and data are communicated in unclear ways that make the findings difficult to understand and interpret. More detail regarding methodologies would

greatly enhance the readability of the manuscript (see examples under Specific comment 1).

Reply: *We now better explain the methodology used in conjunction to the relative sections in the results part of the manuscript. We have also included illustrations that explains the rationale for all optogenetic manipulations included in the manuscript.*

2) The statement that "a similar number of granules approached the plasma membrane in both control and Sac2D cells" is crucial for the interpretation of the entire paper, as it is a basis for concluding that the reduction in docking is the major consequence of Sac2D depletion. The experiment measuring the number of granules approaching must be presented.

Reply: *We thank the reviewer for pointing this out. We agree that the way the docking data was presented leaved open the possibility that what we interpreted as a docking defect could instead indicate a defect in granule trafficking. We have now re-analyzed all docking experiments and the data is presented in absolute numbers of granules approaching the plasma membrane for a given area. The results are presented in **Figure 5E** and show that the same number of granules approach the membrane in both control and Sac2 KD cells, but whereas around 50% of the granules stably dock in control cells, this number is reduced to less than 10% in the Sac2 KD cells.*

3) The requirement of Sac2 activity may not be exclusive to insulin granules. Example: PMID: 2803504 Role of Sac2 in decreasing PI(4)P levels as the phagosome seals in macrophages.

Reply: *We now cite the paper by Levin et al in both the introduction and when discussing how known roles of Sac2 in membrane trafficking relates to our findings.*

"Sac2/INPP5F is a recently characterized, predominantly neuronal, PI(4)P phosphatase that localizes to endosomes and participate in endosome maturation, receptor recycling and phagocytosis (Hsu et al., 2015; Levin et al., 2017; Nakatsu et al., 2015)."

"Sac2 was previously shown to control phosphoinositide turnover in the endocytic compartment, regulating recycling of integrins and transferrin receptors as well as phagocytosis (Hsu et al., 2015; Levin et al., 2017; Nakatsu et al., 2015)."

4) Single puncta (granules?) are shown in various cases (Fig 1E, Fig 2 I-K, Fig 3A). It is not possible to assess whether this is reflective of the population that exists within a cell. It would be preferable to include whole-cell images with multiple magnified ROI's containing individual puncta to illustrate these points, in order to more firmly support the derived conclusions. In Fig 1E, the point is to show that Rab3 is peripheral and NPY is luminal. This is not convincingly conveyed. In addition, Rab3A is not only on insulin granules, this must be clarified, and the data interpreted accordingly.

Reply: *We believe that this comment relates to a misunderstanding on how the data was presented. We apologize for not clearly explaining what the granule images represented. The images of the granules are not examples but average projections from the indicated number of cells (20 granules from each cell). To underscore that these images represent the population average, we have now also included intensity line profiles (means±S.E.M.) that are shown next to the images of the granules, and included a more thorough description on*

how these were generated in the materials and methods section. In addition, we have also included example images of whole cells showing the colocalization between: NPY-mCherry and GFP-Rab5a, NPY-mCherry and GFP-Rab4, NPY-mCherry and GFP-Rab10, NPY-mCherry and LAMP1-GFP, mCherry-Sac2(DN) and GFP-Rab27a, mCherry-Sac2(DN) and NPY-GFP, mCherry-Sac2(DN) and GFP-Rab5a, GFP-Sac2(DN) and mCherry-Sac2 (**Suppl. Fig. 2**), NPY-mCherry and GFP-LactC2 (control and Sac2 KD; **Figure 6E**), NPY-mCherry and GFP-2xFYVE (control and Sac2-KD; **Figure 6A**), NPY-mCherry and GFP-Sac2(DN) (control, wortmannin, PAO, PIK93 and PI4K2A/B KD; **suppl. Fig. 3**), GFP-Granuphilin and NPY-mCherry (control and Sac2-KD, **Suppl. Fig. 5D**). With regards to the comment of luminal vs peripheral granule signal; we do not believe that we can accurately distinguish between these two locations with the resolution of the imaging setups that we have, and this was also never our intention. We have now removed the description “luminal” from NPY-mCherry in the text, not to emphasize this.

Regarding the comment that Rab3A also labels non-granular compartments: We are aware of this. However, a large majority of cellular Rab3a is selectively bound to insulin granules. As can be seen in the picture below, most granules (marked with NPY-mCherry) are Rab3a positive (see also Figure 4 in PMID:24835618, where it is shown that the overlap is around 85%). In our colocalization analysis, we see very similar degree of colocalization between Sac2 and all markers of insulin granules (Rab3a, Rab27a and NPY), so we believe that the Rab3a compartment that we investigate represent the insulin granules and not another compartment (e.g. lysosomes).

5) Does constitutive secretion, mentioned in the Introduction, also require prior depletion of PI(4)P from granules?

Reply: We would also like to know if the proposed role of PI(4)P in constitutive secretion in yeast is conserved in mammalian cells. However, the focus of the current study was on regulated secretion and we feel that studies on the constitutive pathway would be beyond the scope of this study, but certainly represent an interesting future research area.

6) Very few of cells were scrutinized in most instances and this requires clarification:
Examples: Figure 1F: < 10 cells; Figure 2C,D: 8-12 cells; Figure 2F: 7-11 cells; Figure 2H: 6-7 cells; Figure J: 6 cells; Figure 2K,L: 11 cells

Reply: The reason for the low number of cells for certain experimental conditions is that we are only able to visualize 1-2 transfected cells per experiment. This is especially true for the optogenetic experiments which require 100X objective and co-transfection of at least 3 plasmids in a cell line that is difficult to efficiently transfect. Nevertheless, we have now repeated experiments in order to increase the number of cells analyzed.

Figure 1H,I: mCh-Sac2 and GFP-Rab27a (225 structures, 12 cells)
mCh-Sac2 and NPY-GFP (249 structures, 14 cells)
mCh-Sac2 and GFP-Rab3a (212 structures, 11 cells)
mCh-Sac2 and LAMP1-GFP (198 structures, 11 cells)

Figure 2F-G: DMSO (19 cells)
PIK93 (26 cells)
PAO (23 cells)
Wortmannin (26 cells)

Figure 2H,I: Control (33 cells)
PI4K2A KD (36 cells)
PI4K2B KD (39 cells)

Figure 2M: Opto-control (14 cells)
Opto-4'-ptase (16 cells)

Figure 4B,C: Control (87 cells)
Sac2 KD (100 cells)
Sac2-KD + Sac2 (50 cells)
Sac2-KD + Sac2-DN (100 cells)

Figure 4F: Control (27 cells)
Sac2-WT (66 cells)
Sacc2-DN (57 cells)

We have also made sure to match these numbers in all new experiments performed during the course of the revision.

7) It would be important to detect the localization of the endogenous Sac2 on insulin granules (by colocalization with insulin or growth hormone) as well as to detect PI(4) on purified granules. Moreover, some discussion is required as to whether the endogenous levels of Sac2 are limiting, given that overexpressing Sac2-GFP enhanced granule fusion.

Reply: We have attempted to detect endogenous Sac2 by both immunofluorescence and Western blot, but have not succeeded using commercially available antibodies (as has also been reported in previous publications: 25869669 and 25869668). We agree that it is important to confirm the presence of PI(4)P on insulin granules, however we lack this expertise ourselves and were therefore unable to perform such experiments within the timeframe given for revision. Our basis for concluding that the insulin granules contain PI(4)P are:

1) The granules are derived from the trans-Golgi membrane, which is rich in PI(4)P. This lipid is also part of the mechanism that generate secretory granules, so its presence in the granule membrane is expected.

2) We observe changes in insulin secretion when granule PI(4)P is dephosphorylated using the well-established PI(4)P-selective 4'-phosphatase domain from Sac1 (Figure 1D).

3) We observe slight, but significant, enrichment of the PI(4)P-binding PH-domain from OSBP on the granule surface, and the binding is increased when Sac2 expression levels are reduced.

Regarding the overexpression of Sac2 we thank the reviewer for commenting on this. In fact, in Figure 4F we show that overexpression of mCherry-Sac2(DN) results in suppressed depolarization-induced secretion when compared to cells overexpressing wildtype mCherry-Sac2. However, this panel was missing control cells expressing only mCherry. We have now included the control group, which shows that overexpression of wildtype Sac2 has a modest effect on secretion by itself. The kinetics of VAMP2-pHluorin insertion into the plasma membrane seems to be accelerated, but this could be due to either enhanced exocytosis or suppressed endocytosis. In Figure 7C we show that overexpression of wildtype Sac2 increase the number of docked granules, and this is consistent with data from human islets showing that glucose-stimulated insulin secretion positively correlated with Sac2 mRNA levels. A likely explanation for the discrepancy between the docking and secretion data is that the experimental conditions were different: docking is glucose dependent, so in order to maximally promote docking in the experiments presented in Figure 7C these were performed in 20 mM glucose combined with 100 μ M diazoxide to prevent depolarization. The secretion measurements performed in Figure 4 using VAMP2-pHluorin on the other hand, were performed in a basal buffer containing 3 mM glucose. It is therefore possible that glucose-derived signals required for docking are rate-limiting under these conditions, which might explain why we don't observe enhanced secretion.

Specific Comments:

1) Examples of lack of clarity: Supplemental Figure 1A-C: the authors validate that each of the phosphoinositide-modifying constructs used had impact on plasma membrane phosphoinositides. There is a key detail missing from this figure legend and the in-text explanation. In contrast to the results Figure 1A-C where the constructs were targeted to the granular membrane, Supplemental Figure 1A-C instead uses a plasma membrane targeted system that is NOT described (in text or in the figure legend) to verify that each of the constructs was with effect on the plasma membrane. This is distinct from the immediately subsequent Supplemental Figure D-G, where the authors again use the granular membrane-targeted constructs to show that they are without effect on plasma membrane phosphoinositides. Given that these two figures are showing seemingly opposite results (A-C having an effect on PM phosphoinositides, and D-G without effect on PM phosphoinositides), it is crucial to clearly specify these inferred details for clarity and understandability.

Reply: *We apologies for the lack of clarity. We have addressed this by providing graphical illustrations showing the principle of all optogenetic manipulations performed in this study.*

2) Figure 1C: Overexpression of a granule-targeted Sac1 potentiated glucose-stimulated growth hormone secretion. It is essential to show that expression of a granule-targeted Sac2 phosphatase has the same effect.

Reply: *The conclusion from figure 1C is that acute dephosphorylation of PI(4)P on the granule surface by the 4'-phosphatase domain of Sac1 enhances glucose-stimulated GH secretion from MIN6 cells. In figure 2B and supplementary figure 1C we show that the isolated 4'-phosphatase domains of Sac2 and Sac1, respectively, have similar impact on plasma membrane PI(4)P levels, measured as their ability to displace the PI(4)P sensor mCh-P4M_{SidM} from this compartment. We used the 4'-phosphatase domain of Sac1 because it is better characterized than that of Sac2. It would have been interesting to recruit the full-length phosphatases, however this was not possible in the case of Sac2 since CRY2-fusions of the*

full-length protein showed localization to endosomes and granules in the absence of illumination (similar to the untagged full-length Sac2).

3) Figure 1D-F: The text states "we expressed GFP-tagged Sac2 in MIN6 B-cells and imaged the cells by both TIRF and confocal microscopy", but these data are not shown. Instead, the authors present data of a phosphatase-dead DN Sac2 mutant (Fig 1D) and study the localization of this mutant with markers of insulin granules and early and late endosomes (1E,F). Clarification regarding the choice of the DN Sac2 mutant (over the WT) would be helpful. Examples: Figure 1E: images of single granules show co-localization of Sac2 with: Rab27a, Rab3a, and NPY. However, the authors report in Figure 1F that Sac2 co-localizes with additional markers (not shown in 1E): EEA1, Rab5, Lamp1. Given that images for Sac2 co-localization with either early endosome marker were not shown and that Sac2 shows the highest enrichment in this compartment, images are relevant. Figure 1E: y axis 'Fold enrichment' - it is unclear exactly which parameters were used to measure Fold enrichment (not clarified in text or figure caption).

Reply: *We have addressed these points as follows:*

1) *We have added an illustration showing high degree of overlap between wildtype and catalytically inactive Sac2 (suppl. Fig. 2A). In addition, the text has been changed to: "A previous study demonstrated that inactivation of the 4'-phosphatase domain of Sac2 resulted in a stronger punctate distribution (Hsu et al., 2015). In line with this, a phosphatase-dead version of Sac2 (mCh-Sac2DN) showed more pronounced punctate localization that overlapped with wild-type Sac2 and the early endosomal marker GFP-Rab5 as well as markers of insulin granules (GFP-Rab3a, GFP-Rab27a and NPY-GFP). Very little overlap between mCh-Sac2DN and the lysosomal marker GFP-LAMP1 was observed (Fig. 1D-F and Suppl. Fig. 2)."*

2) *See reply to major comment 4 above. We have also added pictures of whole cells to illustrate the degree of overlap between Sac2 and proteins of the endo- and exocytic pathways (Suppl. Fig. 2A). We now also give a more detailed description on how calculations of fold enrichment were performed:*

*"For each cell, a minimum of 15 structures were randomly selected and an average projection image was generated. The line-profiles of these images were measured and normalized to the minimal background fluorescence value (min). To estimate the enrichment of Sac2 and other proteins at the granule site, we first determined the width of the granule (N_{gr}). Next, we integrated the area under the curve for the line profiles of Sac2 (or other proteins) (AUC_{gr}) and the whole range (AUC_{All}). Enrichment was calculated as $[AUC_{gr} - N_{gr} * min] / [AUC_{All} - N_{All} * min]$. If there is no enrichment of Sac2 at the granule site, this value should approach N_{gr} / N_{All} ." See also illustration below:*

4) Figure 2F: It is concluded that PI4P synthesis is not required for Sac2 localization to the insulin granule since inhibition of PI4K did not alter Sac2 enrichment on the granular membrane. However, the data appear to trend towards an involvement of PI4K/PI4P synthesis, given that PAO treatment trends towards decreasing Sac2 enrichment in insulin granules. This in conjunction with only 11 cells tested, make it premature to conclude that there is no role for PI4K in Sac2 targeting to insulin granules.

Reply: See also reply to comment 1 by reviewer 1. We have repeated these experiments and more than doubled the number of analyzed cells. As the reviewer points out, there was a trend towards reduced association between Sac2-DN and insulin granules following incubation with the type III α PI4-kinase inhibitor PAO. This trend now reaches significance (**new figure 2F,G,J**). However, treatment with the broad type III PI4-kinase inhibitor Wortmannin or the type III β PI4-kinase-selective inhibitor PIK93 did not significantly affect the association between Sac2-DN and insulin granules (although there is a tendency to reduced association). Moreover, we have also per recommendation of reviewer 1 reduced the expression level of class II PI4-kinase (PI4K2A and PI4K2B), which was without effect on Sac2-DN binding to insulin granules. We also repeated the optogenetic experiments involving recruitment of a 4'-phosphatase to the granule surface, and can confirm that this results in displacement of Sac2-DN. Together, these experiments show a requirement of PI(4)P for Sac2 binding to insulin granules. In light of the results obtained with PAO, we can no longer conclude that this occurs completely independent of PI4-kinase activity, and we have therefore made the following changes in the text:

“Wortmannin (10 μ M), a non-specific type III PI4-kinase inhibitor, had little effect on GFP-Sac2(DN) binding to insulin granules. Similar results were obtained with the type III β PI4-kinase -selective inhibitor PIK93 (30 μ M). PAO (50 μ M), which primarily targets type III α PI4-kinase, did partially prevent GFP-Sac2(DN) binding to the granules (**Fig. 2F,G,J and Suppl. Fig. 3**). Type II PI4-kinases can localize to vesicular membranes and are insensitive to pharmacological inhibition (Balla et al., 2002; Wiedemann et al., 1996). To test if these isoforms are required for the granular localization of Sac2, we reduced the expression of both type II α and II β PI4-kinases (PI4K2A/B) with siRNA. However, reduced expression of type II

PI4-kinase was without effect on GFP-Sac2(DN)-binding to insulin granules (Fig. 2H-J and Suppl. Fig. XX). Together, these results indicate that the association of Sac2 with insulin granules is largely independent of PI4-kinase activity."

5) Figure 2G&H: Inconsistent treatment group notation. CRY2-rptase and opto-4'-ptase are two different notations used to refer to the same construct in a single figure.

Reply: *CRY2-4ptase refers to the fusion protein in Figures 2L and 2N, whereas opto-4ptase refers to cells co-expressing CIBN-Rab3a and CRY2-4ptase (Figures 2M,O). We have added an illustration (Figure 2K) that will hopefully be helpful when interpreting the data.*

6) Figure 3E: Inconsistent treatment group notation. Knockdown cells are referred to as "siRNA #2", whereas in the remaining panels of the figure, the notation Sac2 KD is used.

Reply: *We thank the reviewer for bringing our attention to this. We now use siSac2 to indicate transient knockdown and shSac2 to indicate stable knockdown.*

7) Figure 5A: This illustration is depicting 'the principle for local, acute stimulation of exocytosis from single cells', however more detail is required in text.

Reply: *We have now added the following text to the figure legend:*

"See materials and methods for details."

And the following description in the section Fluorescence microscopy:

"Exocytosis was evoked with 75 mM K⁺ (equimolarly replacing Na⁺) applied by computer-timed local pressure ejection through a pulled glass capillary."

8) Figure 5C states that Sac2 KD cells have reduced vesicle density at the plasma membrane. Referencing Figure 5E the text states that "a similar number of granules approached the plasma membrane in both control and Sac2D cells", but the data are not shown. Given the clear reduction in vesicle density at the plasma membrane in Sac2 KD cells (Fig 5C), the data in Figure 5E should be revised to report the absolute number of vesicles approaching the plasma membrane (currently reported as %).

Reply: *See reply to Major comment 2 above.*

9) Figure 5E: Does expression of WT-Sac2 recover the defects in granule docking observed in Sac2 KD? Similar data has been shown for Rab3A and Rab27A (both of which are reduced in T2D islets). The authors should consider experiments to prove that the impairment in docking is specifically due to the loss of Sac2 phosphatase activity, and not an auxiliary unknown action of Sac2 that could also contribute to vesicle docking independent of PI(4)P levels. Overexpression of granule-targeted Sac1 (used in Figure 1C) could be used in Sac2 KD cells to restore phosphatase activity against PI(4)P and determine whether or not this reverses impairments in docking could be identified.

Reply: *We attempted to perform the experiments suggested by the reviewer where we recruit the 4'-phosphatase domain of Sac1 to insulin granules in cells with reduced Sac2 expression levels. Although we did observe efficient recruitment of the phosphatase to the surface of granules, we were unable to detect any increase in docking rate during the duration of the experiment. However, because accurate detection of docking requires sampling at high frame rates, significant bleaching occurred that precluded imaging for longer time periods than 10*

min. It is possible that transport of granules to the plasma membrane takes longer time. Still, to substantiate the claim that it is the 4'-phosphatase activity of Sac2 that is important for granule docking, we have repeated the VAMP2-pHluorin experiment in cells with reduced Sac2 expression following attempted rescue with both wildtype and catalytically inactive Sac2. As can now be seen in new **Figure 4B and 4C**, the depolarization-induced exocytosis is rescued by wildtype, but not phosphatase-dead, Sac2. This together with the observations that the granule density at the plasma membrane could be rescued by wildtype, but not phosphatase-dead Sac2 (**Figure 5C and 5D**), are strong evidences that granule docking depends on the 4'-phosphatase activity of Sac2.

10) Supplemental Figure 4: It is shown that docking proteins Rab3a and Rab27a are not lost from granules in Sac2 KD cells, but other factors must be considered, such as SNAP25 and Syntaxin-1.

Reply: We agree with the reviewer that there are numerous potential factors to consider. We have now rephrased the discussions accordingly:

“Rab27 and Rab3 are the best characterized docking factors in β -cells (Kasai et al., 2005; Yaekura et al., 2003), but their localization to granules did not require Sac2. Similarly, the granule localization of the Rab27 effector Granuphilin was also Sac2 independent (Yi et al., 2002). However, other docking factors, including Rabphilin, Exophilin8, Noc2 and Rim2 α , remain interesting candidates that may act downstream of Sac2. Loss of these results in reduced granule docking and release, and the proteins are equipped with a FYVE-like domain that might enable binding to PIs (Cheviet et al., 2004; Fan et al., 2017; Tsuboi and Fukuda, 2005; Yasuda et al., 2010).”

Reviewer #3 (Comments to the Authors (Required)):

Defining mechanisms for how membranes are remodeled within the secretory pathway and how this links to protein trafficking events is an important endeavor. Here, the authors have discovered a role for Sac2 in regulating PI4P levels on insulin granules and provide convincing evidence for the functional significance of this regulation in cultured cells. Some very nice tools for optogenetic erasure of PI4P from insulin granules and measuring exocytosis using pHluorin fusions were developed and used for this study. Live cell imaging data also implicated granule docking at the plasma membrane as being the primary step requiring PI4P degradation by Sac2. The primary conclusions are supported well by the data. The primary weakness of the paper is the lack of mechanistic insight into how PI4P is regulating granule docking. It can be very challenging to identify the relevant effectors of a phosphoinositide, but reasonable candidates could be assessed. In this case, OSBP and cholesterol have also been linked to insulin granule maturation (PMID:29540530). One predicted consequence of Sac2 depletion and excess PI4P accumulation in the granules would be excess cholesterol loading driven by OSBP-mediated PI4P/cholesterol exchange (PMID:28978670).

1. Would it be possible for the authors to test whether OSBP is the relevant PI4P effector that is perturbing granule docking in Sac2 deficient cells? The citations provided above should at least be introduced and discussed in the context of the current work.

Reply: This is indeed an intriguing possibility that we had not thought about. From the previous literature (which is now cited – see below), OSBP acts upstream of cholesterol accumulation, and loss of OSBP results in impaired insulin secretion as a consequence of lysosomal degradation of newly synthesized granules (Hussein, et al, 2018). We measured total insulin content in control cells and cells with reduced Sac2 expression, but did not observe any difference (**Figure 3E**), which would suggest that insulin degradation is unaffected in these cells. However, this specific function of OSBP may be partly independent of cholesterol, since OSBP knockdown could not be rescued by cholesterol supplementation (Hussein, et al, 2018). Cholesterol overloading of insulin granules indeed seems impact insulin granule biogenesis and is associated with both defect docking (Bogan et al, 2012) and exocytosis (Xu et al, 2017), but the connection to PI(4)P is not clear. We believe that the potential involvement of Sac2 in the regulation of granule cholesterol levels warrants further investigation, but we believe that the amount of experimentation required is beyond the scope of the current study. We do, however, now extensively discuss this as a possible explanation for the observed docking defect and a focus of future research.

“An interesting, still unexplored, possibility is that granule PI(4)P is required for cholesterol accumulation in the granule membrane. Insulin granules are rich in cholesterol, and both cholesterol deficiency and overloading results in defect insulin granule biogenesis and secretion (Bogan et al., 2012; Hussain et al., 2018; Xu et al., 2017). Oxysterol binding protein (OSBP) exchanges ER-derived cholesterol for Golgi PI(4)P (Mesmin et al., 2017), and loss of OSBP has been shown to inhibit insulin secretion by preventing the granules from accumulating cholesterol (Hussain et al., 2018). We find that the PH-domain of OSBP weakly binds to the insulin granules, an interaction that is enhanced in cells with reduced Sac2 levels. This observation is consistent with OSBP or related proteins acting as PI(4)P sensors on the insulin granule surface. Interestingly, such a mechanism would resemble the proposed model for how secretory vesicle PI(4)P regulate constitutive secretion in yeast (Ling et al., 2014).”

2. There are a number of typos in the manuscript and descriptions of data in the text were often insufficient to accurately communicate how the experiments were done. For example, OCRL and iSH2 in Fig 1C should be better defined in the results section.

Reply: We apologies for insufficient description of experimental procedures. We have now changed the description in the text relating to Figure 1C to the following:

“Next, we assessed the impact of recruiting the 4'-phosphatase domain of Sac1 (PI(4)P → PI), the 5'-phosphatase domain of OCRL (PI(4,5)P₂ → PI(4)P) or the iSH2-domain of the p85 regulatory subunit of PI3-kinase (PI(4,5)P₂ → PIP₃) on insulin secretion, using growth hormone (GH) as a proxy for insulin to selectively measure release from transfected cells.”

We have also included a box in Figure 1C that shows the enzymatic reaction catalyzed by the respective module.

3. In Figure 2E and other comparable graph the Y axis should be better defined. F/F₀ would not generate a negative value. Isn't this F₀-F?

Reply: The reviewer is correct – the y-axis was incorrectly labeled in Fig. 2B-E and Suppl. Fig. 1. We have now changed the axis to show F/F₀ (so that a reduction in fluorescence below the initial fluorescence [F₀] takes values 0.0 > F < 1.0. In figure 2E, data is presented as (F-

F0)/F0 to specifically emphasize that there is a reduction in plasma membrane associated fluorescence.

4. Supplemental Fig 1 inappropriately truncates the Y-axis to exaggerate effects. The GRP1 designation in B is different from what is in the legend, which uses PH-Akt.

Reply: *The ranges of the y-axis in all panels of figure 1 are scaled to clearly show the kinetics of the respective response. The scale will be different for different biosensors since they all have different dynamic ranges. We do not believe that it is possible to directly compare the relative dissociation of the PH-domain of PLC δ_1 in response to recruitment of 5-ptase $_{OCRL}$ with the relative dissociation of the P4M domain of SidM in response to 4-ptase $_{Sac1}$ recruitment. The magnitude of this response will be affected by the affinity of the biosensor for the respective lipid, the relative abundance of the lipid, the efficiency of the phosphatase, and the expression level of the phosphatase together with the membrane-anchored binding partner. The purpose of this figure was not to show how effective the dephosphorylation or phosphorylation of specific phosphoinositides are, but to show that it is specific for a given phosphoinositide. We are grateful to the reviewer for noticing the inappropriate labeling of panel B. We have now corrected it: panel B shows results obtained with the PH-domain from GRP1 whereas panel G shows results obtained with the PH-domain from Akt1.*

5. Lastly, there is a mouse knockout of Sac2 and I don't see any indication that there is hypoinsulinemia in the mice, although it could be that no one has looked. I think it would be beyond the scope of the current work, but future studies should address the physiological significance of these observations.

Reply: *We thank the reviewer for this suggestion and also for acknowledging that performing these experiments are beyond the scope of this study. We are also not aware of any study specifically addressing the impact of loss of Sac2 on blood glucose regulation. The most extensive study used Sac2 KO mice up to the age of 9 months and investigated the impact on cardiac hypertrophy (19875726). They did not measure blood glucose, and would most likely have missed hyperglycaemia unless it was very severe. In addition, diabetes-like phenotypes often present late in life and many times require other factors, such as high-caloric or high-fat diet. Interestingly, two studies (26908121 and 19875726) demonstrate that Sac2 function as a negative regulator of PI3-kinase, where loss of Sac2 was associated with elevated PI(3,4,5)P $_3$ levels. This is most likely not a direct consequence of Sac2 depletion, but rather reflect the ability of Sac2 to interact with 5'-phosphatases (25869668). Given the importance of PI3-kinase and PI(3,4,5)P $_3$ in the systemic insulin response, it is possible that the β -cell secretory defect that occurs as a consequence of reduced Sac2 expression is partially compensated in insulin target tissues by increased PI3-kinase signalling. Therefore, we believe that in vivo experiments aimed at determining the physiological relevance of Sac2 for blood glucose homeostasis would be best performed in β -cell-specific Sac2 knockout animals.*

July 11, 2019

RE: JCB Manuscript #201903121R

Dr. Olof Idevall-Hagren
Uppsala University, Dept. Medical Cell Biology
Husargatan 3
BMC
Uppsala 75123
Sweden

Dear Dr. Idevall-Hagren,

Thank you for submitting your revised manuscript entitled "The PI(4)P phosphatase Sac2 controls insulin granule docking and release". You will see that all the original reviewers appreciated the revision efforts and are supportive of publication. Rev#1 suggested a number of text revisions to reflect the open questions and highlight surprising results in the paper. Their last point could be dealt with in future work. Rev#3 had several other minor comments that could be addressed without new experimentation. Rev#2 brought up several unresolved questions as well - including endogenous Sac2 distribution studies (which we appreciate has been a challenge for other groups as well and would not be needed for publication) and detecting PI(4) on purified granules (which if not possible could be explained further). Please respond to the remaining questions from the reviewers and discuss these limits. We would be happy to publish your paper in JCB pending these revisions and final revisions necessary to meet our formatting guidelines (see details below). Please include a point-by-point response to the remaining reviewer comments stressing what changes were made to the manuscript at resubmission.

- 1) Text limits: Character count for Articles and Tools is < 40,000, not including spaces. Count includes title page, abstract, introduction, results, discussion, acknowledgments, and figure legends. Count does not include materials and methods, references, tables, or supplemental legends.
- 2) Figure formatting: Scale bars must be present on all microscopy images, including inset magnifications. Please add scale bars to 1G (magnifications), 1H, 2FHOP, 3AC, 5E, 6AE (including magnifications) 6CFGHI, S1EG (including magnifications), S2 (all magnifications), S3AC (all magnifications), S4B, S5ABDE (magnifications included)
Molecular weight or nucleic acid size markers must be included on all gel electrophoresis. Please add molecular weight with unit labels on the following panels: S4D
- 3) Statistical analysis: Error bars on graphic representations of numerical data must be clearly described in the figure legend. The number of independent data points (n) represented in a graph must be indicated in the legend. Statistical methods should be explained in full in the materials and methods. For figures presenting pooled data the statistical measure should be defined in the figure legends.
Please indicate n/sample size/how many experiments the data are representative of: 2J, 4H (please

clarify in the figure legend whether the numbers at the bottom of the bars indicate n), S3B

4) Materials and methods: Should be comprehensive and not simply reference a previous publication for details on how an experiment was performed. Please provide full descriptions in the text for readers who may not have access to referenced manuscripts.

- Please be sure to describe the basic features of all constructs, even if described in other published works/gifts from other researchers - alternatively, please include database IDs (e.g., from Addgene)
- Please include the sequences for all negative controls for sh/siRNAs, if provided to you from the manufacturer, and all shRNA sequences used
- Please include the species and ID/catalog number for all antibodies
- More information about RNA sequencing and gene expression analysis protocols, even if described in other work.
- Microscope image acquisition: The following information must be provided about the acquisition and processing of images:
 - a. Make and model of microscope
 - b. Type, magnification, and numerical aperture of the objective lenses
 - c. Temperature
 - d. imaging medium
 - e. Fluorochromes
 - f. Camera make and model
 - g. Acquisition software
 - h. Any software used for image processing subsequent to data acquisition. Please include details and types of operations involved (e.g., type of deconvolution, 3D reconstitutions, surface or volume rendering, gamma adjustments, etc.).

5) References: There is no limit to the number of references cited in a manuscript. References should be cited parenthetically in the text by author and year of publication.

- Please abbreviate the names of journals according to PubMed.

6) A summary paragraph of all supplemental material should appear at the end of the Materials and methods section.

- Please add ~1 descriptive sentence/item.

A. MANUSCRIPT ORGANIZATION AND FORMATTING:

Full guidelines are available on our Instructions for Authors page, <http://jcb.rupress.org/submission-guidelines#revised>. **Submission of a paper that does not conform to JCB guidelines will delay the acceptance of your manuscript.**

B. FINAL FILES:

-- High-resolution figure and video files: See our detailed guidelines for preparing your production-ready images, <http://jcb.rupress.org/fig-vid-guidelines>.

Thank you for this interesting contribution, we look forward to publishing your paper in the Journal of Cell Biology.

Sincerely,

Vivek Malhotra, PhD
Monitoring Editor, Journal of Cell Biology

Melina Casadio, PhD
Senior Scientific Editor, Journal of Cell Biology

Reviewer #1 (Comments to the Authors (Required)):

The revised manuscript by Nguyen and colleagues is somewhat enigmatic; finding Sac2 on insulin granules is interesting but the role of Sac2 in insulin secretion is still a mystery. A weakness of the study remains the lack of mechanistic insight into how PI4P metabolism regulates granule docking. Also, the nature of the PI4P conversion event remains unknown. PI4P to PI? PI4P to PS? PI4P to cholesterol? The authors have performed the experiments requested by the reviewers, but the results have not revealed a specific role for Sac2 in insulin secretion.

However, this study should not necessarily have to suffer double jeopardy by another review process. The findings that Sac2 is localized to insulin granules and is required for glucose-stimulated insulin secretion are important on their own. As such, this manuscript should be considered for publication pending some minor revisions.

Specific comments:

1. The authors speculate on Sac2 mechanisms in the Discussion. They mention OSBP function, which is plausible. However, the new data provided seem to eliminate a PI4P to PI3P conversion step. It is unclear then why the authors would invoke PI3P-binding proteins in the Discussion. The

following section should either be revised or removed from the manuscript.

"Rab27 and Rab3 are the best characterized docking factors in β -cells (Kasai et al., 2005; Yaekura et al., 2003), but their localization to granules did not require Sac2. Similarly, the granule localization of the Rab27 effector Granophilin was also Sac2 independent (Yi et al., 2002). However, other docking factors, including Rabphilin, Exophilin8, Noc2 and Rim2 α , remain interesting candidates that may act downstream of Sac2. Loss of these results in reduced granule docking and release, and the proteins are equipped with a FYVE-like domain that might enable binding to PIs (Cheviet et al., 2004; Fan et al., 2017; Tsuboi and Fukuda, 2005; Yasuda et al., 2010)."

2. A role for PI 4-kinase III α in generating PI4P on secretory granules is highly unexpected. The authors may want to specifically state that while type II PI4Ks do not seem to be involved, the PI4K isoform that generates PI4P on Sac2-positive vesicles has not been identified in the current study.

3. In the Introduction, the authors may want to revise the text describing roles for PI4P metabolism during secretory vesicle transport and maturation in yeast. The authors state, "In yeast, PI(4)P plays a crucial role in vesicle maturation by recruiting the RabGEF Sec2p that in turn activates the Rab-GTPase Sec4 and promotes myosin dependent granule transport (Mizuno-Yamasaki et al., 2010). The latter step requires removal of PI(4)P, and in yeast this depends on interactions between PI(4)P and the lipid transport protein Osh4p (Ling et al., 2014)".

However, myosin-dependent secretory vesicle transport is PI4P-dependent (Santiago-Tirado et al., 2011). In addition, PI4P removal is proposed to promote Sec2-Sec15 interactions (Mizuno-Yamasaki et al., 2010), not myosin-directed vesicle transport as currently stated in the text. The authors may want to state, "In yeast, PI(4)P plays a crucial role in vesicle maturation by promoting myosin dependent secretory vesicle transport (Santiago-Tirado et al., 2011) and by recruiting the RabGEF Sec2p that in turn activates the Rab-GTPase Sec4 and binds the exocyst component Sec15 (Mizuno-Yamasaki et al., 2010). The latter step requires removal of PI(4)P, and in yeast this depends on interactions between PI(4)P and the lipid transport protein Osh4p (Ling et al., 2014)".

4. Based on the very relevant previous work by the Bretscher and Novick labs in yeast cells, the authors may want to examine whether Sac2 activity is involved in similar conserved functions. For example, does Sac2 recruitment result in release of a motor protein from insulin granules? This step may be necessary for subsequent docking/tethering events at the plasma membrane. However, this may be beyond the scope of the current study.

Reviewer #2 (Comments to the Authors (Required)):

Many of the comments were attended to in a highly satisfactory manner, and the manuscript is much improved and appears compelling.

A few points were not addressed, however, and the authors might be willing to explain why this was not possible or consider addressing some of them, as follows:

- Detecting the localization of the endogenous Sac2 on insulin granules
- Detecting PI(4) on purified granules
- Fig 1D-F: need to show the data listed as not shown or give citation if done elsewhere, that "we expressed GFP-tagged Sac2 in MIN6 B-cells and observed numerous small punctate structures

that were either static or dynamic"

- Figure 2G&H: the labeling needs to be uniformized: it still appears as CRY2-4ptase in Figure 2L and opto-4'-pase in 2M

Reviewer #3 (Comments to the Authors (Required)):

This manuscript provides strong evidence for a role of Sac2 and PI4P in insulin granule exocytosis and this is a significant advance for the field. A weakness of the manuscript is a lack of mechanistic insight into how PI4P influences granule docking and the authors have not provided further mechanistic insight in this revision. However, I am willing to accept the author's argument that to fully test effector candidates (like OSBP and sterol loading) would add substantial length to a manuscript that is data-rich in its current form. Most of my other concerns were minor and the authors have made the requested corrections.

There are still a few minor issues with the paper that the authors could address.

1. End of Introduction - Type 2 diabetics should be diabetes.
2. Figure 6, F and G. This should probably read ...control (F) and Sac2 KD (G) cells.
3. Our institutional Responsible Conduct of Research instructors claim it is inappropriate to truncate the Y-axis of a figure to exaggerate the differences observed. If this is also JCB's policy, it seems Fig 2J and perhaps 2M should show the full data range from 0.

Reviewer #1 (Comments to the Authors (Required)):

The revised manuscript by Nguyen and colleagues is somewhat enigmatic; finding Sac2 on insulin granules is interesting but the role of Sac2 in insulin secretion is still a mystery. A weakness of the study remains the lack of mechanistic insight into how PI4P metabolism regulates granule docking. Also, the nature of the PI4P conversion event remains unknown. PI4P to PI? PI4P to PS? PI4P to cholesterol? The authors have performed the experiments requested by the reviewers, but the results have not revealed a specific role for Sac2 in insulin secretion.

However, this study should not necessarily have to suffer double jeopardy by another review process. The findings that Sac2 is localized to insulin granules and is required for glucose-stimulated insulin secretion are important on their own. As such, this manuscript should be considered for publication pending some minor revisions.

Specific comments:

1. The authors speculate on Sac2 mechanisms in the Discussion. They mention OSBP function, which is plausible. However, the new data provided seem to eliminate a PI4P to PI3P conversion step. It is unclear then why the authors would invoke PI3P-binding proteins in the Discussion. The following section should either be revised or removed from the manuscript.

"Rab27 and Rab3 are the best characterized docking factors in β -cells (Kasai et al., 2005; Yaekura et al., 2003), but their localization to granules did not require Sac2. Similarly, the granule localization of the Rab27 effector Granuphilin was also Sac2 independent (Yi et al., 2002). However, other docking factors, including Rabphilin, Exophilin8, Noc2 and Rim2 α , remain interesting candidates that may act downstream of Sac2. Loss of these results in reduced granule docking and release, and the proteins are equipped with a FYVE-like domain that might enable binding to PIs (Cheviet et al., 2004; Fan et al., 2017; Tsuboi and Fukuda, 2005; Yasuda et al., 2010)."

Reply: We have revised this part of the discussion, which now reads:

Rab27 and Rab3 are the best characterized docking factors in β -cells (Kasai et al., 2005; Yaekura et al., 2003), but their localization to granules did not require Sac2. The Rab27 effector Granuphilin is also required for insulin granule docking (Gomi et al., 2005), but its localization to insulin granules was also Sac2 independent. However, other docking factors, including Rabphilin, Exophilin8, Noc2 and Rim2 α , remain interesting candidates that may act downstream of Sac2 (Cheviet et al., 2004; Fan et al., 2017; Tsuboi and Fukuda, 2005; Yasuda et al., 2010). These exocytic factors are equipped with FYVE-like domains similar to the PI(3)P-binding FYVE-domains of many endocytic factors. Although PI(3)P has been found on the surface of secretory granules (Dominguez et al., 2011; Wen et al., 2008), we were unable to detect any colocalization between the PI(3)P sensor 2xFYVE^{Hrs1} and insulin granules, and acute dephosphorylation of PI(4)P on the granule surface did not result in accumulation of the PI(3)P sensor. It is possible that the PI(3)P sensor used here, which is from an endocytic factor, fails to recognize the granule pool of PI(3)P. It is also possible that the FYVE-domains of the exocytic factors binds to other PIs, as it has been found that not all FYVE-domains are PI(3)P selective (Gil et al., 2012).

2. A role for PI 4-kinase IIIalpha in generating PI4P on secretory granules is highly unexpected. The authors may want to specifically state that while type II PI4Ks do not seem to be involved, the PI4K isoform that generates PI4P on Sac2-positive vesicles has not been identified in the current study.

Reply: We have now added the following sentences to the discussion:

"Pharmacological inhibition or siRNA-mediated silencing of PI4-kinases partially prevented the association of Sac2 with insulin granules, indicating that part of the granule PI(4)P is synthesized after budding from the Golgi. The specific PI4-kinase isoforms involved remains to be determined."

3. In the Introduction, the authors may want to revise the text describing roles for PI4P metabolism during secretory vesicle transport and maturation in yeast. The authors state, "In yeast, PI(4)P plays a crucial role in vesicle maturation by recruiting the RabGEF Sec2p that in turn activates the Rab-GTPase Sec4 and promotes myosin dependent granule transport (Mizuno-Yamasaki et al., 2010). The latter step requires removal of PI(4)P, and in yeast this depends on interactions between PI(4)P and the lipid transport protein Osh4p (Ling et al., 2014)".

However, myosin-dependent secretory vesicle transport is PI4P-dependent (Santiago-Tirado et al., 2011). In addition, PI4P removal is proposed to promote Sec2-Sec15 interactions (Mizuno-Yamasaki et al., 2010), not myosin-directed vesicle transport as currently stated in the text. The authors may want to state, "In yeast, PI(4)P plays a crucial role in vesicle maturation by promoting myosin dependent secretory vesicle transport (Santiago-Tirado et al., 2011) and by recruiting the RabGEF Sec2p that in turn activates the Rab-GTPase Sec4 and binds the exocyst component Sec15 (Mizuno-Yamasaki et al., 2010). The latter step requires removal of PI(4)P, and in yeast this depends on interactions between PI(4)P and the lipid transport protein Osh4p (Ling et al., 2014)".

Reply: We thank the reviewer for noticing this mistake. We have now corrected the text in the introduction accordingly.

4. Based on the very relevant previous work by the Bretscher and Novick labs in yeast cells, the authors may want to examine whether Sac2 activity is involved in similar conserved functions. For example, does Sac2 recruitment result in release of a motor protein from insulin granules? This step may be necessary for subsequent docking/tethering events at the plasma membrane. However, this may be beyond the scope of the current study.

Reply: We thank the reviewer for this suggestion. We did not observe any difference in insulin granule movement in Sac2 knockdown cells, and the number of granules that approached the plasma membrane was similar to control cells. This would indicate that myosin-dependent granule transport occurs independent of Sac2. However, it would be interesting to investigate, as the reviewer suggests, whether there is a defect in the disassembly of the motor protein as the granule reaches the plasma membrane. However, we believe that this is beyond the scope of the current study.

Reviewer #2 (Comments to the Authors (Required)):

Many of the comments were attended to in a highly satisfactory manner, and the manuscript is much improved and appears compelling.

A few points were not addressed, however, and the authors might be willing to explain why this was not possible or consider addressing some of them, as follows:

- Detecting the localization of the endogenous Sac2 on insulin granules

Reply: We are aware of this limitation with the study. We have performed immunofluorescence analysis with numerous antibodies directed against Sac2 (PA5-21562, ThermoFischer Scientific; SAB2700848, SigmaAldrich; ab95990, Abcam) but none of them produced detectable signal in our cell line. Similar problems have been reported previously (PMID: 25869669, 25869668).

- Detecting PI(4) on purified granules

Reply: We agree that it is important to confirm the presence of PI(4)P on insulin granules, however we lack this expertise ourselves and were therefore unable to perform such experiments within the timeframe given for revision. Our basis for concluding that the insulin granules contain PI(4)P are:

- 1) The granules are derived from the trans-Golgi membrane, which is rich in PI(4)P. This lipid is also part of the budding mechanism, so its presence in the granule membrane is expected.
- 2) We observe changes in insulin secretion when granule PI(4)P is dephosphorylated using the well-established PI(4)P-selective 4'-phosphatase domain from Sac1 (Figure 1D).
- 3) We observe slight, but significant, enrichment of the PI(4)P-binding PH-domain from OSBP on the granule surface, and the binding is increased when Sac2 expression levels are reduced.

- Fig 1D-F: need to show the data listed as not shown or give citation if done elsewhere, that "we expressed GFP-tagged Sac2 in MIN6 B-cells and observed numerous small punctate structures that were either static or dynamic"

Reply: We have now added a new supplementary figure (S2A) that shows the presence of both static and dynamic GFP-Sac2 positive structures.

- Figure 2G&H: the labeling needs to be uniformized: it still appears as CRY2-4ptase in Figure 2L and opto-4'-pase in 2M

Reply: We apologize for the unclarity. CRY2-4ptase indicate the fluorescent protein, whereas opto-4'ptase indicate the combination of CIBN-CAAX and CRY2-4ptase. We have now added "GFP-" in front of CRY2-4ptase in figures 2L and 2N to better indicate that this.

Reviewer #3 (Comments to the Authors (Required)):

This manuscript provides strong evidence for a role of Sac2 and PI4P in insulin granule exocytosis and this is a significant advance for the field. A weakness of the manuscript is a lack of mechanistic insight into how PI4P influences granule docking and the authors have not provided further mechanistic insight in this revision. However, I am willing to accept the author's argument that to fully test effector candidates (like OSBP and sterol loading) would add substantial length to a manuscript that is data-rich in its current form. Most of my other concerns were minor and the authors have made the requested corrections.

There are still a few minor issues with the paper that the authors could address.

1. End of Introduction - Type 2 diabetics should be diabetes.

Reply: Thank you. This is now corrected.

2. Figure 6, F and G. This should probably read ...control (F) and Sac2 KD (G) cells.

Reply: The reviewer is correct. This has now been changed.

3. Our institutional Responsible Conduct of Research instructors claim it is inappropriate to truncate the Y-axis of a figure to exaggerate the differences observed. If this is also JCB's policy, it seems Fig 2J and perhaps 2M should show the full data range from 0.

Reply: We agree with the reviewer that truncations should be avoided. However, in the case of figure 2J, calculations of protein enrichment at the granule site was performed using the following method:
*To estimate the enrichment of Sac2 and other proteins at the granule site, we first determined the width of the granule (N_{gr}). Next, we integrated the area under the curve for the line profiles of Sac2 (or other proteins) (AUC_{gr}) and the whole range (AUC_{All}). Enrichment was calculated as $[AUC_{gr} - N_{gr} * min] / [AUC_{All} - N_{All} * min]$. If there is no enrichment of Sac2 at the granule site, this value should approach N_{gr} / N_{All} . N_{gr} / N_{All} will approach 0.38 when there is no enrichment, and the axis has been truncated at this value (which is essentially the true zero). If JCB policy requires us to change the y-axis of figure 2M we would be happy to do so.*